# Omics-Based Investigations of Breast Cancer

**DOI:** 10.3390/molecules28124768

**Published:** 2023-06-14

**Authors:** Anca-Narcisa Neagu, Danielle Whitham, Pathea Bruno, Hailey Morrissiey, Celeste A. Darie, Costel C. Darie

**Affiliations:** 1Laboratory of Animal Histology, Faculty of Biology, “Alexandru Ioan Cuza” University of Iasi, Carol I Bvd, No. 20A, 700505 Iasi, Romania; 2Biochemistry & Proteomics Laboratories, Department of Chemistry and Biomolecular Science, Clarkson University, 8 Clarkson Avenue, Potsdam, NY 13699, USA; whithad@clarkson.edu (D.W.); brunop@clarkson.edu (P.B.); morrisha@clarkson.edu (H.M.); celestedarie@gmail.com (C.A.D.)

**Keywords:** breast cancer (BC), genomics, transcriptomics, proteomics, metabolomics, new omics, onco-breastomics

## Abstract

Breast cancer (BC) is characterized by an extensive genotypic and phenotypic heterogeneity. In-depth investigations into the molecular bases of BC phenotypes, carcinogenesis, progression, and metastasis are necessary for accurate diagnoses, prognoses, and therapy assessments in predictive, precision, and personalized oncology. This review discusses both classic as well as several novel omics fields that are involved or should be used in modern BC investigations, which may be integrated as a holistic term, onco-breastomics. Rapid and recent advances in molecular profiling strategies and analytical techniques based on high-throughput sequencing and mass spectrometry (MS) development have generated large-scale multi-omics datasets, mainly emerging from the three ”big omics”, based on the central dogma of molecular biology: genomics, transcriptomics, and proteomics. Metabolomics-based approaches also reflect the dynamic response of BC cells to genetic modifications. Interactomics promotes a holistic view in BC research by constructing and characterizing protein–protein interaction (PPI) networks that provide a novel hypothesis for the pathophysiological processes involved in BC progression and subtyping. The emergence of new omics- and epiomics-based multidimensional approaches provide opportunities to gain insights into BC heterogeneity and its underlying mechanisms. The three main epiomics fields (epigenomics, epitranscriptomics, and epiproteomics) are focused on the epigenetic DNA changes, RNAs modifications, and posttranslational modifications (PTMs) affecting protein functions for an in-depth understanding of cancer cell proliferation, migration, and invasion. Novel omics fields, such as epichaperomics or epimetabolomics, could investigate the modifications in the interactome induced by stressors and provide PPI changes, as well as in metabolites, as drivers of BC-causing phenotypes. Over the last years, several proteomics-derived omics, such as matrisomics, exosomics, secretomics, kinomics, phosphoproteomics, or immunomics, provided valuable data for a deep understanding of dysregulated pathways in BC cells and their tumor microenvironment (TME) or tumor immune microenvironment (TIMW). Most of these omics datasets are still assessed individually using distinct approches and do not generate the desired and expected global-integrative knowledge with applications in clinical diagnostics. However, several hyphenated omics approaches, such as proteo-genomics, proteo-transcriptomics, and phosphoproteomics-exosomics are useful for the identification of putative BC biomarkers and therapeutic targets. To develop non-invasive diagnostic tests and to discover new biomarkers for BC, classic and novel omics-based strategies allow for significant advances in blood/plasma-based omics. Salivaomics, urinomics, and milkomics appear as integrative omics that may develop a high potential for early and non-invasive diagnoses in BC. Thus, the analysis of the tumor circulome is considered a novel frontier in liquid biopsy. Omics-based investigations have applications in BC modeling, as well as accurate BC classification and subtype characterization. The future in omics-based investigations of BC may be also focused on multi-omics single-cell analyses.

## 1. Introduction

Breast cancer (BC) is known as a “diverse collection of neoplastic diseases” [1]. In-depth investigations into the molecular bases of BC phenotypes, development, progression, and metastasis are required for accurate diagnoses, prognoses, and assessments of the response to therapies for improving long-term patient survival [2]. The term “omics” gathers various fields of study that explore patients’ bodies at different molecular levels, ranging from the genome and transcriptome to the proteome and metabolome [3]. Moreover, interactomics promotes a holistic view in BC research by constructing and characterizing the protein–protein interaction (PPI) networks that provide a novel hypothesis for the pathophysiological processes involved in BC progression and subtyping. Recently, three “epi-omics” fields, epigenomics, epitranscriptomics, and epiproteomics, have emerged, in order to explore the modifications that occur on the DNA, RNA, and proteins, respectively, for an advanced understanding of the cellular structure and function [3]. Novel omics fields, such as epichaperomics and epimetabolomics, could investigate the modifications in the interactome induced by cellular stressors and provide proteome-wide changes in PPI [4] and metabolites removed from their classical function, known as epimetabolites [5], respectively, as drivers of BC-causing phenotypes. However, each omics approach is centered only on one dimension of a biological system, while multi-omics strategies offer multidimensional perspectives on extremely complex biological and pathological processes. In the omics-data integration era of predictive, precision, and personalized medicine [6], based on patient-centered approaches [7], the use of integrative molecular studies has emerged via the high-throughput profiling of the DNA, RNA, protein, PTMs, PPIs, and metabolite level, which is is mandatory in order to explain the complexity of this heterogeneous, malignant phenomena [8]. In recent years, many dry-lab and wet-lab based studies have identified large sets of molecular biomarkers for BC that already were or are still waiting to be further integrated by bioinformatics, statistical approaches [9], and artificial intelligence [10], resulting in systems biology comprehensive analyses [11] and network-based approaches that are applicable in clinical trials [2,7]. Web applications that provide rapid analyses of multi-omics data sets from a significant number of primary breast tumors, i. e., Cancer Target Gene Screening (CTGS), have been developed [12].

Cancer-specific online omics databases and clinical databases for BC research that collect patient-related, tumor-related, diagnostic-related, treatment-related, outcome-related, administration-related, and other clinical data that are publicly available [13] are excellent sources for novel biomarker discovery and molecular profiling in BC [14]. Hence, rapid and recent advances in molecular profiling technologies have generated large-scale biomolecular cancer multi-omics data, known as “big data” [10], emerging from the “four big omics” (genomics [15], transcriptomics, proteomics, and metabolomics) that unveiled novel faces of breast cancer biology at multiple levels of the omics interaction network [16]. Multi-scale systems biology approaches investigate breast cancer through the large-scale quantification of numerous biomolecules [17] and integrate these large volumes of omics data in order to understand this disease from genes to protein–protein interactions (PPIs) and from proteoforms to dysregulated metabolic pathways, for patient stratification, therapy assessments, and prognoses [16]. Gene expression profiling using DNA microarrays [15] has significantly contributed to the understanding of the molecular heterogeneity of BC formation, progression, and recurrence. Today, large platforms for biomarker analyses use high-throughput approaches in genomics, transcriptomics, proteomics, metabolomics, and bioinformatics to more completely describe the biological interactions within a living system during various diseases [18]. However, most of these data are still assessed individually using distinct approaches and do not generate the desired and expected global-integrative knowledge with applications in clinical diagnostics [19]. Nevertheless, comprehensive genomic, transcriptomic, and proteomic profiling may act together to demonstrate the susceptibility, drivers, therapeutic responses, resistance to treatment [20], and intertumoral heterogeneity in metastatic BC [21], the specific molecular landscape in male BC [22], the landscape of novel peptides from coding and noncoding sequences mapped to cancer hallmark genes in BC [23], or new distinct BC subtypes, such as the immune and hormone-related subtypes of invasive lobular carcinoma (ILC) [24]. Each omics-based method has different ways in which it could offer clinical benefits. Omics-based methods are able to do much more than just accurate BC classification. For example, there are discovery experiments in proteomics, such as biomarkers, but one can also look at the metabolome, PTMs of proteins, or PPI networks to determine what is globally happening in a patient’s body. Another relatively new method is to investigate the pharmacokinetics and pharmacodynamics of protein-based drugs in the blood stream and at the tumor level (tumor microenvironment). Administering a protein drug (i.e., antibodies or protein therapeutics) requires its quantification in the blood over the duration of the treatment, which is performed using targeted quantitative proteomics. There is no better and more precise method than targeted proteomics. The only alternative option is ELISA, which can be specific if the antibodies are good, but we know that most of the antibodies on the marker can be less specific than they should be or even unspecific. Therefore, yes, targeted mass-spectrometry-based quantitative proteomics or metabolomics are the best options for the future, with multiomics to follow.

Several hyphenated omics approaches, such as proteo-genomics [25], proteo-transcriptomics [26], and phosphoproteomics-exosomics [27], are useful for the identification of putative BC biomarkers, BC molecular subtypes, and novel therapeutic targets. To develop non-invasive diagnostic tests and discover new biomarkers for BC, classic omics-based strategies have allowed significant advances in blood/plasma-based omics. Salivaomics [28], urinomics, and milkomics have appeared as integrative omics that may develop a high potential for early diagnoses in BC. Thus, an analysis of the tumor circulome is considered a novel frontier in liquid biopsies. Omics-based investigations have applications in BC modeling, as well as BC classification and subtype characterization. The future in omics-based investigations of BC may be also focused on multi-omics single-cell analyses using single-cell-based advanced technologies [29].

## 2. Breast Cancer Investigation in the Multi-Omics Era

According to the central dogma of molecular biology, the genetic information codified in the DNA (genome) is transcribed into RNA (transcriptome), translated into proteins (proteome), and finally also results in metabolites (metabolome) [30] (Table 1). Both DNA and histone proteins are reversibly modified (epigenome), while an analogous process takes place for RNA (epitranscriptome), as well as for proteins (epiproteome) [31]. Consequently, multiple types of molecular data are becoming available for the same set of clinical samples [32], such as the new concepts raised so far. At a monogene level, according to the old concept of “one-gene/one-protein/one-function” [4] at a time that considered a gene function in isolation from its interacting partners, the inherited mutations in *BRCA1* and *BRCA2*, the strongest susceptibility tumor suppressor genes for BC, among other cancers [33], have longtime been considered as the main genetic factors in BC [34]. The advances in high-throughput technologies have switched the old genetics-based approaches that interrogated individual variants or single genes to novel, revolutionary genomics-based approaches focused on the study of entire genomes [35]. In this context, the study of the genomics basis of BC cancer became more complex, involving other molecular layers, such as the transcriptome, translatome, proteome, and interactome [36]. Hence, genomics have identified mutations in a plethora of BC candidate genes and genomics-based biomarker testing has detected molecular variations in single genes, panels of genes, or entire genomes [37]. Then, both genomics- and transcriptomics-based studies have led to the characterization of novel genome-driven integrated classifications of BC that define the integrative clusters associated with distinct clinical outcomes, specific molecular drivers, and oncogenic pathways [38,39,40,41]. Additionally, genomics is directly involved in dissecting BC tumor heterogeneity and is also applied to the treatment of HER2-overexpression and triple-negative BC [42].

High-throughput sequencing technologies have revolutionized the fields of genomics and transcriptomics and, consequently, the medical research that benefits from multi-omics approaches to disease [35,43]. Parallel DNA and RNA sequencing approaches have generated large-scale data on thousands of BC genomes [44]. However, genomics-based diagnostic rates are limited at aproximately 50% across various Mendelian diseases [45]. Single-gene mutations with a high penetrance have been analzyed using genome-wide association studies (GWAS), which explore hundreds of thousands of genetic variants across human genomes to find those significantly associated with a specific feature or disease and to detect the genetic risk factors prevalent in a target population [46,47,48]. GWAS has identified more that 200 susceptibility loci for BC [49] and the discovery of these novel BC susceptibility loci has provided a better understanding of genetic predispozitions and increased subtype-specific risks to BC in genetically isolated populations, such as Ashkenazi Jewish [50], Arab [51], or European populations [52]. A recent study based on transcriptome-informed genome-wide gene-environment interaction suggested a limited role of gene–environment interactions in BC risk [49]. However, these studies are useful for understanding the interactions between the environmental risk factors and genetic variants in BC, as in the case of the pro-inflammatory signaling and gene-lifestyle interaction [53]. Both GWAS and transcriptome-wide association studies (TWAS) have been also performed for the identification of the novel loci associated with mammographic density phenotypes [54] and for the characterization of the multiple signaling pathways associated with BC development in women of Asian and European descent [55]. RNA intereference (RNAi) pathway/gene silencing-based studies aim to correlate the RNAi categories, such as microRNA (miRNA) and small interfering RNAs (siRNA), to different BC types and stages in comparison to healthy cells, emphasizing their high diagnostic, monitoring, and therapeutic potentials [56]. Thus, the post-translational regulatory process, mediated by miRNA and siRNAs, which prevents gene expression in cancer, as well as in other diseases [57], is widely involved in the elucidation of gene function [58]. Even if the tumor genome and transcriptome are important tools for the discovery of novel biomarkers for BC, the dysregulated proteome expression reflects more accurately the essential changes in the tumor pathophysiology [59]. High-throughput mass-spectrometry (MS)-based techniques enable more comprehensive insights into changes in the proteome to advance personalized medicine [59]. Moreover, GWAS have emphasized the complementarity of proteomics to RNA-seq in capturing the functional impact of rare genetic variations [45]. To avoid invasive tumor tissue biopsies or surgeries, over the last decades, various omics-based strategies have allowed for significant advances in searching for non-invasive or minimally-invasive biomarkers for all-stage and especially early-stage BC diagnoses in cancer liquid biopsies. Blood/plasma-based genomics, usually consisting of cell-free DNA (cfDNA) or circulating tumor DNA (ctDNA) analyses, are useful for diagnoses in BC patients and the prediction of disease-free survival (DFS) [60], assessments of triple-negative breast cancer (TNBC) progression and personalized management [61], sub-clonal diversification in advanced BC [62], BC pre-diagnosis [63], and dormancy [64]. Blood-based epigenomics studies suggest the possibility of using blood-based DNA methylation markers as a promising tool for BC risk stratification [65]. A plethora of plasma-based transcriptomics studies are deeply involved in biomarker discovery for BC diagnoses, using extracellular vesicle long RNA-sequencing (exLR-seq) [66], plasma-derived extracellular vesicle circular RNAs-sequencing (circRNA-seq), and qRT-PCRs [67,68], as well as circulating micro-RNAs qRT-PCRs [69]. LC-ESI-MS/MS-based techniques are able to identify plasma peptides and phosphopeptides to differentiate BC from other diseases [70]. Metabolomics-based approaches are useful for identifying new diagnostic biomarkers for BC [71]. Hence, targeted plasma-based metabolomics, using liquid chromatography-mass spectrometry (LC-MS) [71], as well as liquid chromatography-tandem mass spectrometry (LC-MS/MS), for metabolic profiling have identified metabolic candidate biomarkers that enable a high sensitivity and the specific detection of all-stage and early-stage BC [72]. Untargeted liquid chromatography quadrupole time-of-flight mass spectrometry (LC-QToF-MS) and targeted LC-triple quadrupole mass spectrometry (LC-QQQ-MS) have compared the plasma metabolite profiles in BC patients to those from healthy controls, emphasizing the potentially deregulated pathways that contribute to BC pathogenicity [73]. Integrative analyses of plasma metabolomics and proteomics have revealed specific changes in the metabolic and proteomic profiling of BC patients, emphasizing sphingomyelins, glutamate, and cysteine as potential diagnostic biomarkers for BC [74]. LC-MS/MS-based proteomics help to identify the tumor subtype-specific biomarkers within BC interstitial fluid [75]. Lipidomics-based studies have analyzed the correspondence between the lipid BC features observed by both desorption electrospray ionization-mass spectrometry imaging (DESI-MSI) in tissue and those detected using LC-MS in the plasma of BC patients [76].

To develop non-invasive diagnostic tests, genomics-, epigenomics-, proteomics-, and metabolomics-based approaches may be used to identify the cellular changes associated with the precancerous and early stages of BC through nonaggressive sampling, even before the cell alterations become detectable using biopsy/histopathological analyses [77]. In patients with ductal carcinoma in situ (DCIS) stage I, an epigenomics-based noninvasive analysis of nipple aspirate fluid (NAF) unveiled the hypermethylation of one or more genes that were absent in the benign and normal breast tissue, as well as in the NAF, from healthy women [78]. Thus, promoter hypermethylation has been reported as a promising biomarker for BC. Also, the detection, isolation, and characterization of breast-tumor-derived components from saliva may be used for multi-omics examinations of BC patients [79]. Saliva may be considered as a promising, efficient, and noninvasive source of protein biomarkers and other dysregulated proteins, emphasizing a high potential to accurately differentiate BC patients from healthy controls [80]. As it preserves good-quality genomic DNA [81], saliva was recommended as a potential alternative for detecting hereditary BC mutations using Next Generation Sequencing (NGS) [82]. Targeted proteomics-based analyses have highlighted the peptide-based specific signatures in saliva as putative predictors to distinguish between TNBC and healthy subjects [83]. Untargeted metabolomics-based approaches and bioinformatics have unveiled the salivary metabolite profiles in women with BC, highlighting the up- or downregulated metabolites associated with BC and putative useful biomarkers for BC [84]. To reveal the full mRNA transcriptome and develop a test based on the detection of cell-free RNA from saliva, Bentata et al. showed an enrichment of genes with a functional annotation in alternative splicing that can serve as an indicator for BC [85]. Hence, integrative salivaomics targets the discovery of noninvasive biomarkers and includes salivary genomics [82], epigenomics, transcriptomics [85], proteomics [83], metabolomics [84,86], immunomics, and microbiomics approaches [87]. Also, salivaomics is useful for a determination of the metabolic characteristics of saliva, depending on the molecular biological subtype of BC in correlation with the expression levels of HER2, estrogen receptors (ER), and progesterone receptors (PR) [28]. Urine-based diagnosis techniques are also non-invasive, inexpensive, sensitive, and easy to use in clinical settings [88], reflecting pathological changes, especially in the early stages of the disease [89]. Urinomics is focused on non-invasive biomarkers discovery, including proteomics [90], transcriptomics, exosomics, and metabolomics approaches [91]. Urinary exosomal miRNAs have been proposed as potential noninvasive biomarkers in BC detection [92]. Breast milk analyses also have value in risk detection, early detection, or diagnoses of BC [93]. Thus, milkomics integrates the results of proteomics-based studies that offer promising biomarkers for the early detection of BC [94,95,96]. Also, deregulated circular RNAs (circRNAs) from milk and other liquid biopsies may have clinical relevance as diagnostic, prognostic, and predictive biomarkers that play key roles in breast tumorigenesis and BC progression [97].

**Table 1 molecules-28-04768-t001:** Omics-based investigation of BC and related technologies.

	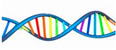	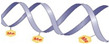	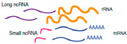	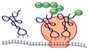	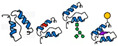	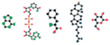
Central dogma of cancer biology	nDNA,cfDNA/ctDNA, mtDNA	aberrant DNA methylation,HMs [98]	mRNA, ncRNAs [99]: circRNAs [100,101], miRNA,snRNA, snoRNA, piRNA, and lncRNA [102]	translating mRNAs, rRNAs [103], tRNAs [104,105], regulatory ncRNAs, nascent polypeptide chains [106]	peptides, proteins, isoforms, proteoforms,protein–protein interaction networks	metaboliteslipids
Omes	genome	methylome	transcriptome	translatome	proteomephosphoproteomeacethylproteomeglycoproteomeinteractome	metabolomelipidome
Omics	Genomics	Epigenomics	TranscriptomicsmiRomics	Translatomics	ProteomicsPhosphoproteomicsGlycoproteomicsInteractomics [107]	Metabolomics [30]Lipidomics
Technologies	DNA microarray [108,109];sc-genomics/scDNA-seq [110];RT-qPCR in tissue [111] and plasma [112];DNA-seq: first generation seq, NGS (WGS [113];WES [114,115], targeted gene sequencing); GWAS [52,116];mtDNA-seq (tissue and NAF [117,118])	sc-epigenomics;microfluidics assays;NGS (single-gene NGS,genome-wide DNA methylation analysis seq,ChIP-seq);MS for HMs; RNA-seq for miRNAs [98]	sc-transcriptomics/scRNA-seq (CITE-seq [119,120]),RNA microarray [121]; microarray-based STRNA qRT-PCR;NGS: RNA transcription group seq (whole transcriptome analysis, snRNA-seq, ncRNAs analysis)	translating RNA (polysome profiling, ribo-seq, RNC-seq, TRAP-seq);tRNAome: (2-DE, MS, HPLC, NGS, Ribo-tRNA-seq);folding state of nascent polypeptides (X-ray diffraction, cryo-EM, NMR);identification and quantification of nascent peptides (pSILAC, BONCAT/QuaNCAT, PUNCH-P);in vivo visualization of translation (FRET)	LC-MSLC-MS/MS [122];LC-ESI-MS/MS [70];MALDI-ToF MS [123];MALDI-ToF-MSI, multiplex MALDI-IHC and LC-MS/MS [124];SELDI-ToF-MS for NAF [125,126];DESI-FAIMS-MSI [127];SP3-CTP multiplex MS proteomics [128]	NMR (LC-NMR and GC-NMR) and MS (LC-MS and GC-MS) [129];GC-ToF MSCE-ToF-MSLC-ESI-MSLC-MS/MSLC-QToF-MS and LC-QQQ-MS [73];RRLC-ESI-MS/MSHR-MAS MRS [121];lipid tissue signatures by DESI-MSI [76];MasSpec Pen [130]

Abbreviations: BONCAT/QuaNCAT—Bio-Orthogonal/Quantitative Non-Canonical Amino acid Tagging; cfDNA—cell free DNA; CITE—seq-Cellular Indexing of Transcriptomes and Epitopes; ChIP—seq-chromatin immunoprecipitation-deep sequencing; circRNAs—circular RNAs; ctDNA—circulating tumor DNA; DESI-FAIMS-MS—desorption electrospray ionization-high field asymmetric waveform ion mobility; EM—electron microscopy; FRET—Fluorescence Resonance Energy Transfer; GC—gas chromatography; GWAS—genome-wide/whole genome association studies; HMs—histone modifications; HR MAS MRS—high resolution magic angle spinning magnetic resonance spectroscopy; LC—liquid chromatography; miRNS—microRNA; mtDNA—mitochondrial DNA; NAF—nipple aspirate fluid; ncRNAs—non-coding RNAs; nDNS—nuclear DNA; NGS—Next-Generation Sequencing; NMR—Nuclear Magnetic Resonance spectroscopy; piRNA—Piwi-interacting RNA; pSILAC—pulsed-Stable Isotope Labelling with Amino Acids in Cell Culture; PUNCH-P—PUromycin-associated Nascent Chain Proteomics; Ribo-seq-ribosome profiling; rRNAs—ribosomal RNAs; RNC-seq—Ribosome Nascent-Chain complex sequencing/full-length translating mRNA sequencing; scRNA—seq-single-cell RNA sequencing; snRNA—small nuclear RNA; snoRNA—small nucleolar RNA; ST—spatial transcriptomics; tRNAs—transfer RNAs; TRAP-seq—Translating Ribosome Affinity Purification sequencing; and WES—whole exome sequencing.

Each omics has their own analysis tools to aid in providing more comprehensive information. For example, in the proteomics-based domain, we can determine the gene names that encode for the identified proteins, which can then be used for gene set enrichment analyses (GSEA) and search tools for the retrieval of interacting genes/proteins (STRING) analyses. This is just one example of proteomics experiments, but each omics field has different tools to obtain a more comprehensive analysis of their data. Some of the tools/platforms available online are listed below:BigOmics Analytics (https://www.bigomics.ch, accessed on 7 June 2023), in which the company has an easy to use set of tools called “Omics Analysis for Everyone—Easy-to-use omics tool”;BioCyc (https://biocyc.org/omics.shtml, accessed on 7 June 2023) offers omics data analyses. The website offers multiple tools for the analysis of gene expression, metabolomics, and other large-scale datasets. Options for gene expression and metabolomics data are detailed here, but many of the options that involve pathways or the metabolic map can also be used for proteomics, multi-omics, or other kinds of high-throughput data;NetGestalt (https://www.altexsoft.com/blog/omics-data-analysis/, accessed on 7 June 2023) is a web app for multi-omics data visualization and integration;MiBiOmics (https://shiny-bird.univ-nantes.fr/app/Mibiomics, on 7 June 2023) is an interactive web-based (and standalone) application for easily and dynamically exploring associations across omics datasets;Subio Platform (https://www.subioplatform.com/, accessed on 7 June 2023) is professional software for analyzing quantitative omics data such as transcriptomics, epigenetics, or proteomics.

Hence, there are many options for the analysis of multiomics datasets and many of them have the potential to provide more comprehensive information. When choosing an omics-based method, we first need to think of the outcome needed. Are we looking for determining the proteome, metabolome, genome, or other “omes”? Each omics method has differing amounts of time from the sample preparation to the analysis, as well as pitfalls. Genomic sequencing using DNA microarrays, RT-qPCR, and DNA-seq all have their own sets of pros and cons. Microarrays have a high sample throughput and can be relatively cost effective, but only can identify known genes and transcripts. Proteomics-based methods would be able to identify all the expressed proteins in the proteome and identify which proteins are dysregulated based on specific conditions, but can be time consuming due to the instrument time. Multi-omics and single-omics methods are significantly more expensive than others due to their specificity. Multi-omics combines multiple omics-based approaches to give a comprehensive understanding of the molecular changes that can contribute to the disease state, cellular response, and development. Single-cell-omics methods are specialized to reflect only one aspect of a biological system at a single-cell resolution, but they are not feasible for large-scale samples, thus not being a method for looking at multiple disease states. Both single-cell omics and multi-omics methods are high cost, and due to this, small sample sizes are optimal.

All omics-based methods have emphasized their own pros and cons, as well as optimal sample size, sensitivity, and cost. There is no way to compare all of them to determine which is the best omics method to use, as each method tells a different story about what is happening in a patient’s body. The stage of BC really would depend on what we are looking to obtain from the dataset. Are we looking for biomarkers (early-onset BC), or for molecular mechanisms of metastasis (mid- to late-stage BC)? This could decide which BC stages are best for a specific method.

### 2.1. Genomics- and Epigenomics-Based Investigation of Breast Cancer

Genomics, the most mature of the omics fields, is the study of the whole genomes of an organism [35]. First, BC was considered a disease of the genome, with around 15% of all BC patients being genetically predisposed to BC [113]. The somatic mutation theory of carcinogenesis proposed that gene mutations alone cause cancer, while the tissue organization field theory of carcinogenesis and neoplasia showed that cancer is a supracellular phenomena, involving interactions among the cells in the stroma of an organ that appears as the primary target of carcinogenesis [131]. Hence, mutations cannot be considered to be the unique cause for cancer cells to develop; changes in the energy pathways or the intra- and extracellular molecular landscapes are an equal cause of cancer cell initiation, in addition to mutations in the oncogenes and tumor suppressor genes [132]. Genomics has been included in diagnostic medicine as clinical genomics, which offers important information about the genomic drivers of cancer development and progression [133]. Importantly, genomics may be applied even to the treatment of BC, starting from the identification of mutations in a panel of candidate genes but targeting the individual molecular complexity and intraheterogeneity of breast tumors [42].

Today, next generation sequencing (NGS)/high-throughput sequencing/deep-sequencing technology is widely used in gene research and has replaced first-generation sequencing (FGS) due to its high speed, high throughput, and high accuracy [134]. The NGS in BC research is mainly used in genomics-, epigenomics-, transcriptomics-, and translatomics-based investigations, using genome DNA sequence analyses (including whole-genome sequencing (WGS) [113], whole-exome sequencing (WES) [135], and targeted sequencing [136]), RNA transcription group sequencing (RNA-seq) (including whole-transcriptome sequencing (WTS) [137], small RNAs sequencing, including circulating small RNAs, small RNAs in extracellular vesicles [138], and non-coding RNA analyses [139]), and epigenetic sequencing (including chromatin immunoprecipitation followed by sequencing (ChIP-seq) [140] and genome-wide DNA methylation analyses [141], [134]. WES, RNA-seq, and WGS are implemented in routine clinical settings, demonstrating the clinical relevance of genomics in cancer medicine [113]. Epigenetic alterations play critical roles in the pathogenesis of BC [142]. Epigenomics studies the phenotype changes that are independent of a DNA sequence [143]. The epigenetics drivers, such as histone modifications (HMs), including histone acethylation and methylation, as well as DNA methylation and epigenetic regulatory mechanisms based on miRNAs, induce a dysregulation of the genes related to BC cell differentiation, survival, migration, and invasion [98]. HMs are essential players that maintain genomic stability, transcription, and DNA repair and modulate the chromatin in cancer cells [144]. DNA methylation regulates gene expression without changing the gene sequence, emphasizing global hypomethylation and/or gene-specific hypermethylation [98]. Thus, a methylomics-based study used a methylatyed DNA immunoprecipitation microarray (MeDIP-chip) on DNA samples isolated from the white blood cells of young patients with BC, emphasizing the hypermethylation of several genes, such as *APC*, *HDAC1*, *GSK1*, *SLC6A3*, *Rab40C*, *ZNF584*, and *FOXD3* [142]. Additionally, metagenomics analyses of normal and malignant breast microbiomes using 16S rRNA gene sequencing identified specific microbial communities that could serve as potential biomarkers for prognoses and diagnoses, with implications in developing novel strategies for the treatment of BC patients [145].

Mutations in the tumor suppressors and oncogenes encoded by the nuclear DNA (nDNA) are known to play essential roles in breast carcinogenesis [146]. Thus, the breast cancer susceptibility gene 1 (*BRCA1*), breast cancer susceptibility gene 2 (*BRCA2*), phosphatase and tensin homolog (*PTEN*), tumor protein 53 (*TP53*), cadherin 1 (*CDH1*), and serine/threonine kinase 11 (*STK11/LKB1*) are known as highly penetrant genes, while the cell cycle checkpoint kinase 2 (*CHEK2*), *BRCA1* interacting helicase/protein 1 (*BRIP1*), ataxia-telangiectasia mutated gene (*ATM*), and partner and localizer of BRCA2 (*PALB2*) are known as moderately penetrating genes that suffer mutations, confering up to an 80% and 2–3% lifetime risk of BC, respectively [147]. A multi-Patient-Targeted (MPT) single-cell DNA sequencing (scDNA-seq) approach reconstructed mutational lineages and identified early mutational and copy number alterations (CNA) in TNBC tissue samples [110]. Additionally, mitochondrial DNA (mtDNA) germline variants and somatic tumor mutations are also involved in BC development [118], especially those involved in the OXPHOS system [146].

In the rapidly progressing field of genomics, the development and progression of novel assays based on the profiling of DNA isolated from non-/minimally invasive liquid biopsies, such as blood and blood derivatives, urine, sputum, milk, tear fluid, and other body fluids that contain tumor-derived DNA, may develop a high potential [133] and become more attractive in clinical applications [148]. The analysis of circulating tumor-derived components, known as the “tumor circulome”, is useful for assessing the clonal heterogeneity of tumors, unlike tissue biopsies [149], and is considered to be a new frontier of liquid biopsies [150]. The tumor circulome includes circulating tumor nucleic acids (ctNAs: ctDNA and ctRNA), circulating tumor cells (CTCs), tumor-derived extracellular vesicles (tdEVs), and tumor-educated platelets (TEPs) that can be used as cancer biomarkers [149]. Furthermore, plasma circulating proteins play key roles in BC development and represent an important source of BC biomarkers [151]. Analyses of the whole blood gene expression in blood samples have emphasized the potential links between the immune system and metastasis [63]. Based on parallel sequencing and digital genomics techniques, cfDNA analyses in the blood of BC patients have become a promising biomarker in breast oncology [152]. Combined analyses of the *PIK3CA* mutational status in ctDNA, CTCs, and tdEVs using real-time PCRs (RT-PCR) and next generation sequencing (NGS) have shown that the targeting of these genes in HR+/HER2− metastatic BC has significant benefits after the occurrence of endocrine therapy resistance through the modulation of the PI3K/AKT/mTOR signaling pathway [150].

Circulating tumor DNA (ctDNA) represents up to 10% of a patient’s cell-free DNA (cfDNA) [153]. Compared to tumor biopsies, ctDNA, the cell-free fragmented DNA (cfDNA) that originates from tumor cells and is present in the blood plasma, may be a non-invasive way of detecting genomic information from a limited amount of plasma [154]. It is known that PCR-based technologies, as well as genome sequencing, have different clinical utilities in ctDNA analyses in BC [148], including metastatic BC [154]. Thus, RT-PCR, Amplification-Refractory Mutation System (ARMS-PCR), Peptide Nucleic Acid/Locked Nucleic Acid (PNA/LNA), Beads, Emulsion, Amplification, Magnetic digital PCR (BEAMing-PCR), droplet digital PCR (ddPCR), Co-amplification at Lower Denaturation temperature (COLD-PCR), and differential Strand at Critical Temperature (DISSECT) assays are presented as the main techniques involved in PCR-based approaches and may also be useful for investigating liquid biopsies in BC [148]. For example, the polymorphism in miRNA genes may affect the interaction of miRNAs with their target messenger RNAs, as in the case of miRNA146a, which was significantly overexpressed in BC, especially in aggressive BC subtypes, compared to healthy tissue [155]. In the context of PCR-based methods applied to blood-based analyses in BC patients, the cost-effective Tetra-primer ARMS (T-ARMS) approach was developed and optimized for the detection of the *miRNA-146a* gene rs2910164 (C/G) single-nucleotide polymorphism (SNP) [156], as well as the FAS and FASL polymorphisms involved in the dysregulation of the apoptotic pathway [157]. Likewise, ddPCR is an accurate method for quantifying the DNA copy numbers in BC tissues [158] and has also demonstrated its clinical utility as a valuable technique for the mutational profiling of ctDNA in BC patients, especially in the case of *HER2*, *PIK3CA*, *ESR1*, and *TP53*, the most frequently mutated genes in BC [159]. In addition, as a predictive biomarker of immunotherapy efficacy, microsatellite instability (MSI) detection using ddPCR allows for the large pre-screening of BC patients [160]. Plasma-based genotyping via cfDNA is a promising strategy with clinical applications for the detection of *BRCA1/2* somatic mutations to guide putative therapeutic interventions for patients with metastatic BC [161]. Synchronous Coefficient of Drag Alteration (SCODA), Tagged-amplicon deep Sequencing (TamSeq), the Safe Sequencing System (SafeSeqS), Cancer Personalized Profiling by deep Sequencing (CAPP-Seq), and Targetted Error Sequencing (TEC-Seq) are used for targeted deep-sequencing through both NGS and NGS-PCR [148]. The detection of genome-wide rearrangements using WGS and WES technologies is suggested as a novel opportunity for ctDNA profiling [148]. These last two approches include the Personalized Analysis of Rearranged Ends (PARE) and digital karyotyping [148].

Human saliva contains cfDNA originating from the host (70%), as well as from the oral microbiota (30%) [162]. This biofluid sample is easy to collect and may be recommended as an alternative source for the identification, characterization, and validation of hereditary breast cancer germline mutations using NGS [82]. Saliva-exosomics, known as next-generation salivaomics, assures a comprehensive molecular characterization of the salivary exososmes from saliva, becoming a promising approach for the accurate detection of premalignant lesions and early-stage cancers [163]. Genomics-based analyses of urinary ctDNA have emphasized the somatic mutations linked to the primary tumors of BC patients, such as the *PIK3CA* and *TP53* mutated genes, offering the non-invasive probing and real-time monitoring of BC relapse [164]. Based on cfDNA-seq, *NF1*, *CHEK2*, *KMT2C,* and *PTEN* were found to be the most frequently occurring mutated genes in the plasma and urine of BC patients [165].

### 2.2. Transcriptomics- and Translatomics-Based Investigation of Breast Cancer

Transcriptomics is the study of the transcriptome, known as the total complement of coding and non-coding RNA transcripts in a cell at a time [19], which provides key information about gene regulation and cellular protein content [166]. Transcriptomics analyses have become one of the most popular platforms for the identification of BC-causing key genes and PPI that might play key roles in BC diagnoses, prognoses, and therapy [2]. Additionally, spatial transcriptomics (ST) is an in situ capturing method that allows for the quantification and visualization of transcriptomes in individual histological tissue sections, distinguishing non-malignant, ductal carcinoma in situ (DCIS) and invasive ductal carcinoma (IDC) regions in clinical biopsies of the breast using an automatic selection of cell types via their transcriptome profiles [167]. Epitranscriptomics focuses on the understanding of the epitranscriptome, which plays a key role in the alternative splicing, nuclear export, transcript stability, and translation of RNAs [143]. Epitranscriptomic modifications play essential roles in developmental processes and stress responses [31]. The majority of all transcripts comprises non-coding RNAs (ncRNAs), which regulate messenger RNA (mRNA) expression and protein products. Therefore, many genes involved in BC are influenced by ncRNA activity. ncRNAs are divided into two main groups: house-keeping ncRNAs (tRNAs, rRNAs, snRNAs, snoRNAs, and TERC) and regulatory ncRNAs, comprising short ncRNAs (miRNAs, siRNAs, and piRNAs) and long non-coding RNAs (lncRNAs) [102]. The cell-type-specific miRNA expression profile is studied using miRomics, which has demonstrated its clinical utility for the classification of tumor samples and the prediction of prognosis or therapeutic responsiveness [168]. Associated with increased proliferation, a degraded extracellular matrix (ECM), and a higher epithelial-to-mesenchymal transition (EMT), the abnormalities in the functions and expressions of lncRNAs, Piwi-interacting RNAs (piRNAs), small nuclear RNAs (snRNAs), and small nucleolar RNAs (snoRNAs) are involved in the development and progression of BC [102]. Additionally, circular RNAs (circRNAs) may modulate the cell proliferation, migration, apoptosis, and invasion of BC cells in vitro or tumor growth and metastasis in vivo [101]. CircRNA-related translatomics and proteomics are involved in carcinomas and emphasize the therapeutic potential in various diseases [169]. Translatomics investigates all the molecular players involved in the translation process, including translating mRNAs, ribosomes/ribosomal RNAs (rRNAs), transport RNAs (tRNAs), regulatory RNAs, such as miRNA and lncRNA, and nascent polypeptide chains [106]. Some of the main methods of translatomics, such as polysome profiling, full-length translating mRNA profiling (RNC-seq), translating ribosome affinity purification (TRAP-seq), and ribosome profiling (Ribo-seq), as well as tRNA-specific NGS, nuclear magnetic resonance (NMR) spectroscopy, pulsed-Stable Isotope Labelling with Amino Acids in Cell Culture (pSILAC), Bio-Orthogonal/quantitative Non-Canonical Amino acid Tagging (BONCAT/QuaNCAT), and PUromycin-associated Nascent Chain Proteomics (PUNCH-P), have been reviewed by Zhao et al. and are listed in Table 1.

RNA sequencing (RNA-seq) may predict the progression and aggressiveness of BC [170,171]. The transcriptomic profiling of breast tumor samples has emerged as one of the most powerful approaches in holistic oncology over the last decades [172]. This an excellent tool for understanding drug bioactivity, discovering novel molecular targets for anticancer therapies, identifying candidate biomarkers, such as the mRNAs and miRNAs proposed and introduced as biomarkers in BC subtype classification [173], assessing therapy resistance [174], or identifying BC patients who may be spared endocrine treatment [175]. RNA dysregulation resulting in aberrant RNA transcripts translated into tumor-specific proteins is a source of tumor antigens involved in the development of new immuno-therapeutical targets [176]. Co-culturing cell models followed by RNA-seq have emphasized the BC cell transcriptomic response induced by tumor-associated macrophages (TAMs), demonstrating the TAMs impact on almost all the signaling pathways involved in BC tumorigenesis, such as the transcription, translation, molecule transport, and immune-related pathways [177].

miRNAs are key regulatory molecules involved in the post-transcriptional regulation of gene expression via RNA interference (RNAi) through the binding of protein-coding mRNA [168] and are aberrantly expressed in solid tumors, including BC, suggesting their involvement in carcinogenesis [178]. Based on analyses of microRNAs/miRome, miRomics studies are an available source of biomarkers for the diagnosis and treatment of BC [179]. Thus, analyses of the tissue and blood miR-191, miR-22, and EGFR mRNA may play a role in BC prognosis and for obtaining diagnostic biomarkers for patients with BC [180]. MicroRNA signatures can be used as diagnostic, predictive, or prognostic cancer biomarkers for the development of miRNA-based targeted therapeutics [181], with a potential use in the personalized medicine of BC [182]. Cell differentiation, proliferation, apoptosis, metastasis, relapse, and chemoresistance are controlled by either oncogenic miRNAs or tumor-suppressor miRNAs (tsmiRNAs) [183]. Thus, small RNA sequencing and qRT-PCR analyses have shown that certain miRNA species, such as miR-21 and miR-1246, are selectively enriched in BC exosomes and significantly elevated in the plasma of patients with BC. Thus, they may be considered as promising candidate biomarkers due to their cancer-specific expression profiles [184].

Several tissue-based mRNA tests are routinely used in clinical practice for assessing BC recurrence risk and guiding treatment decisions [185,186]. The study of the presence and expression of the various tumor-associated circulating transcripts (TACTs) in the peripheral blood of BC patients through transcriptomics-based analyses may help to discriminate between controls and patients, to profile BC subtypes, or to decipher the mechanisms involved in BC development and progression. Thus, many potential biomarkers for the diagnosis and evaluation of BC progression have been identified in peripheral blood by assessing the expressions of human mammoglobin A (MAM) mRNA [187], ERBB2/HER2 mRNA [188,189], EPCAM, KRT19, MKI67, TERT, VIM, NPTN, MCAM, SNAI2, and FOXA2 mRNAs [190]. mRNA-based diagnostic blood tests have been developed for the early detection of BC, which are cost-effective and high-throughput [185,190,191]. Furthermore, cytokeratin 19 (CK19) mRNA was identified in the CTCs disseminated in the peripheral blood of patients with early-stage BC [186].

Transcriptomics and proteomics [123], as well as genomics- and proteomics-based approaches in the saliva-exosomics field [163], have demonstrated significant differences between BC patients and controls, leading to the discovery of salivary biomarkers for the detection of BC with a high specificity and sensitivity. ctDNA and ctRNA, including messenger RNA (mRNA), miRNA, and lncRNA as potential biomarkers, are found in saliva, as well as in other bodily fluids [79]. miRomics studies are an available source of salivary biomarkers for the diagnosis and treatment of BC. Thus, the differential abundance of splicing factor mRNA in the saliva is recommended as a diagnostic tool for BC [85]. A miRomics-based study showed that urinary miRNAs, such as the overexpressed miR-155 and underexpressed miR-21, miR-125b, and miR-451, may be proposed as non-invasive and innovative urine-based candidate biomarkers for BC detection [192]. In addition, exosomics-based studies have emphasized a specific panel of urinary microRNAs, including miR-424, miR-423, miR-660, and let7-I, as a highly specific biomarker tool for discriminating BC patients from healthy controls [92].

### 2.3. Proteomics-Based Investigation of Breast Cancer

Even if genomics- and transcriptomics-based classifications and biomarker discovery in BC are still largely used for diagnoses, prognoses, and the prediction of patients’ outcomes, current clinical tests and treatment decisions are more focused on protein-level investigations [128]. Proteomics is the study of the proteome, known as the entire set of proteins in a given cell, tissue, or biological sample, at a precise developmental or cellular phase [19]. Hence, proteomics refers to the large-scale study of proteins, including protein identification, ontology, protein–protein interaction (PPI) networks, post-translational modifications (PTMs), pathway involvment, quantifications, and functional analyses [193]. Untargeted quantitative proteomics is the first step in the identification of the dysregulated proteins in different conditions. However, once this is performed and a list of the dysregulated proteins becomes available, then many of them may become potential biomarkers. The verification of these biomarkers can then be performed also using targeted quantitative proteomics. Targeted proteomics also looks for specific peptide sequences to quantify the amount of proteins within a sample and compare the disease and healthy states. Single-reaction monitoring (SRM) and multi-reaction monitoring (MRM) are two ways of quantifying the protein levels between two samples. SRM only looks for one peptide sequence or only one *m/z*, whereas MRMs look to quantify multiple peptide sequences, therefore scanning over many *m/z* ratios based on the method development. This is just one way of utilizing proteomics-based methods to quantify the protein level in a sample. Moreover, proteomics involves the identification and characterization of protein subgroups, such as kinome (kinomics), secretome (secretomics), exosome (exosomics), degradome (degradomics), and phosphoproteome (phosphoproteomics) [193]. Matrisomics is focused on the study of matrisome, which comprises approximately 300 extracellular matrix (ECM) proteins, a large number of ECM-modifying enzymes, ECM-binding growth factors, and other ECM-associated proteins [194], so that the omics data provide novel insights into the ECM function in the development, homeostasis, and disease condition [195]. A plethora of matrisomics-based methods have been reviewed by Neagu et al., emphasizing the dysregulated proteins involved in ECM/TME remodelling and EMT in invasive ductal carcinoma (IDC) of the breast [196]. Thus, an LC-MS/MS technique was successfully applied for emphasizing the roles of multiple proteins in ECM/TME remodelling, such as tenascin C (TNC) [197], collagen isoforms (COL1A1, COL1A2, and COL14A1), fibronectin 1 (FN1), mimecan/osteoglycin (OGN), decorin (DCN), and thrombospondins [198]. Additionally, MALDI-FT-ICR MS, MALDI-ToF MS, and MALDI tandem mass spectrometry have been applied to interrogate the IDC-dysregulated proteins involved in ECM/TME remodelling [196]. A combined approach using proteomics and single-cell transcriptomics showed how cancer associated fibroblast (CAF)-secreted collagen XII alters collagen I organization to induce a pro-invasive breast tumor microenvironment that supports metastatic dissemination [199]. Exosomes are biomolecular nanostructures released by cells into biological fluids [200]. Tumor cells also release small extracellular vesicles (sEVs), which may contribute to carcinogenesis and tumor progression [201]. Exosomics is known as the study of the exosome, which encompasses an entire set of exosomal proteins [193]. Exosomes are a rich source of BC-related proteins and miRNA that may be used for BC diagnoses and prognoses [202]. Hence, MS-based proteomics and NGS are usually used to investigate exosomes as potential sources for novel BC biomarkers [202]. Thus, the nanoLC-MS/MS technique has been applied to explore the proteomic profile of cancerous and non-tumorigenic breast cell lines [201], while the MALDI-ToF/ToF MS approach was performed for the proteomic profiling of the plasma and total blood exosomes in BC [203]. A phosphoproteomics-exosomics-based investigation used LC-MS/MS to identify the EV phosphoproteins from human plasma as potential biomarkers for differentiating BC patients from healthy controls [27]. In 2000, Tjalsma et al. introduced the term secretome to characterize both the pathways for protein export and the secreted proteins [204]. Today, this definition includes soluble proteins, as well as lipids and EVs, which carry important molecules [205]. However, the term secretomics reflects a proteomics-based approach to identifying and quantifying all the proteins secreted by different cell types through unconventional protein secretion (UPS) [206]. The cancer secretome includes proteins secreted by tumor cells (cytokines and growth factors), ECM/TME proteins, proteases and protease inhibitors, membrane and EVs proteins, peptide hormones, and metabolic proteins [207]. McHenry and Prosperi published a comprehensive review on the proteins found in the TNBC secretome and their therapeutic potential [207].

The reciprocal actions of protein kinases and phosphatases play key roles in many cellular processes [208]. Kinomics is the study of global kinase activity, usually based on peptide microarrays [209]. Quantitative proteomics-based methods, combined with genomic analyses of the mutations and expression in the kinome and the driver oncogenes upstream of specific kinase networks, have been used used to emphasize the dynamic behavior of the cancer kinome [210]. Each BC subtype emphasizes a unique expression profile of protein kinases, which are valuable candidates for new targeted therapies based on kinase inhibition, due to their key roles in tumorigenesis and cancer progression [211,212]. A complex study including an LC-MS/MS analysis elucidated the phosphosites and kinases implicated in TNBC, suggesting a target-based clinical classification for TNBC [213]. The serine/threonine kinase (AKT) signaling pathway is activated via phosphorylation, leading to the cell growth, proliferation, survival, and activation of glucose metabolism, while AKT signaling pathway mutations, especially in *PI3KCA* and *PTEN*, are associated with a resistance to hormonal treatment in BC patients [214]. Moreover, based on the importance of the PTMs that induce structural and functional changes in the proteins involved in multiple biological and pathological processes, the integration of phosphoproteomics data analyses in multi-omics cancer studies becomes challenging [32]. Thus, phosphoproteomics contributes to the identification of key biomarkers for assessing BC pathogenesis and its drug targets [215]. In this context, large-scale, quantitative, MS-based, phosphoproteomics-based studies have emphasized the dynamic landscape of receptor tyrosine kinases (RTKs) [208], which are overexpressed or dysregulated in BC cells and lead to accelerated tumor growth, angiogenesis, and metastasis via the activation of various downstream cell-signaling pathways [216]. To investigate the intrinsic and adaptive mechanisms of resistance to phosphatidylinositol 3-kinase (PI3K) inhibition in TNBC, a complex study based on transcriptomics-, proteomics-, kinomics-, and phosphoproteomics-specific technologies, including LC-MS/MS and MS-based kinome profiling applied to a panel of PDX models of TNBC, revealed the potential roles of serine/threonine-protein kinase NEK9 and mitogen-activated protein kinase kinase 4 (MAP2K4) in mediating buparlisib resistance and emphasized the role of omics-based analyses in unveiling the resistance mechanisms to targeted therapies [217]. In order to emphasize the heterogeneity of TNBC, in-depth, high-throughput phosphoproteomics exhibited certain advantages over several gene-centred approaches, such as targeted NGS and whole-genome array-comparative genomic hybridization (aGCH), methylomics, and transcriptomics [213]. Hence, various phosphoproteomics-based techniques have emerged as powerful tools for biomarker discovery, BC classification, and the prediction of treatment outcomes using clinical samples. Thus, large-scale, differential, phosphoproteomics-based analyses of human BC tissues in high- and low-risk recurrence groups have been performed using immobilized metal affinity chromatography (IMAC), coupled with the isobaric tag for the relative quantification (iTRAQ) technique, followed by strong cation exchange chromatography (SCX), liquid chromatography electrospray ionisation tandem mass spectrometry (LC-ESI-MS/MS) analyses, and subsequent selected/multiple reaction monitoring (SRM/MRM) for validation [215].

Cancer is considered a ”disease of pathway” [218]. To explain how genome-encoded products interact in complex networks, interactomics studies these various interactomes and provides a global view of PPI networks [107]. Protein-network-based approaches result in subneworks that provide a novel hypothesis for the pathways involved in BC progression, as well as in subnetwork biomarkers that are more reproducible and emphasize a higer accuracy in the classification of metastatic vs. non-metastatic BC [218]. Additionally, integrated proteo-transcriptomics analyses have demonstrated the potential of uncovering the novel characteristics of BC, emphasizing that the globally increased protein-mRNA concordance may be associated with tumor proliferation, aggressive BC subtypes, and decreased patient survival [26]. These kinds of hyphenated omics approaches are useful for explaining the origin of the abnormal levels of some proteins in BC patients’ blood. In the case of cytidine deaminase (CDA), an enzyme of the pyrimidine salvage pathway, both the CDA protein activity levels and CDA mRNA levels were higher in the blood samples from BC patients than in those from controls [219].

In recent years, invasive techniques for diagnosing and monitoring cancers have tended to be replaced by non-invasive or minimally-invasive methods, such as liquid biopsies [220]. Novel omics-related terms have been proposed and defined in relation to the various bodily fluids that replace the blood and blood-derivatives that have been the medium of choice for diagnosing most diseases. For example, salivaomics and urinomics are emerging fields that include proteomics-, epigenomics-, metabolomics-, immunomics-, microbiomics-, and transcriptomics-based investigations of various diseases [87]. A plethora of proteomics-based techniques, such as MALDI-ToF MS, SELDI-ToF MS, MALDI-ToF/ToF MS, and LC-MS/MS, have commonly been used to emphasize single biomarkers or panels of serum biomarkers for BC. Moreover, the emerging proteomic results help to discriminate between patients with BC and healthy controls, analyze the proteins involved in the various biological processes and pathways that are dysregulated in BC, and identify stage-specific protein biomarkers or discriminate between the aberrantly externalized proteins produced by BC cells or stromal cancer-associated cells [221]. A study carried out in 2000 showed that a salivary proteomics-based panel of biomarkers, consisting of ERBB2/HER2, cancer antigen 15-3 (CA15-3), and tumor suppressor oncogene protein 53 (*TP53*/p53), has a potential use in the initial detection and follow-up screening for the detection of BC in women based on ELISA detection [222]. Today, isotopic labeling coupled with liquid chromatography tandem mass spectrometry (IL-LC-MS/MS) is often used to characterize the salivary protein profiles in BC patients [223]. Salivary protein biomarkers are known to be useful for discriminating between healthy controls, fibroadenomas and incipient ductal carcinoma lesions [224], and lymph-node-positive and lymph-node-negative patients with ductal carcinoma [225], as well as to assess the differences between the salivary protein profiles of HER2/*neu*-receptor-positive and HER2/neu-receptor-negative patients [226]. A recently published study based on isobaric tag for relative and absolute quantitation (iTRAQ)-based MS/MS proteomics analyses identified the differentially expressed proteins proposed as putative biomarkers for early BC diagnosis and prognosis, which were then evaluated for the construction of PPI interaction networks in the saliva and serum of BC patients [227]. More than 3000 unique proteins have been detected in human urine using MS-based techniques [89]. Proteomics-based approaches, such as LC-MS/MS and MALDI-ToF/ToF MS, may be used to assess the efficacy of method optimization for protein extraction from urine samples [228], for detecting the upregulated, stage-specific, or biomarker proteins involved in the early screening detection and monitoring of invasive BC progression [90], and to identify the significant urinary changes in protein profiles before a breast tumor becomes palpable [89] or the specific urinary proteome alteration in HER2-enriched BC [229].

Taking into account that glycosylation is one of the most important PTMs of a protein, several glycomics- and glycoproteomics-based approaches, which combine immunohistochemical methods, lectin-recognition-based methods, MS-related methods, and fluorescence imaging-based in situ methods, offer a wide potential for discovering and using glycomics and glycoprotein biomarkers in various cancers [230]. Glycomics-based studies have revealed that truncated O-glycans are found in 90% of BCs, while mucin is one of the earliest discovered BC serum biomarkers [230]. The glycoproteomics-based changes in the surface glycoproteins, combined with quantitative proteomics, have been studied in BC under drug treatment [230]. Salivary glycopaterns have been reported as potential biomarkers for the screening of early-stage BC [231]. Quantitative glycoproteomics using LC-MS/MS analyses applied to two highly invasive BC cell lines have provided a comprehensive list of the core-fucosylated glycoproteins involved in the signaling networks that drive BC progression [232]. The relevance of several studies that have used proteomics-based and proteomics-derived omics are listed in Table 2.

### 2.4. Metabolomics-Based Investigation of Breast Cancer

Metabolic alterations are a hallmark of tumorigenesis [84]. Metabolomics is focused on the analysis of the low-molecular-weight metabolites in the metabolome that reflect the dynamic response to genetic modifications [30]. Salivary metabolomics based on LC-MS analyses has shown potential for exploring new salivary biomarkers for discriminating BC patients from healthy controls [80,84,236], while capillary electrophoresis time-of-flight mass spectrometry (CE-ToF-MS) is useful for conducting a saliva metabolomics-based study for the discrimination between oral-, breast-, and pancreatic cancer-specific profiles [237]. Analyzing the metabolic features of saliva in BC patients, it has been shown that the changes in the activity of metabolic enzymes were more pronounced in ductal carcinoma compared to lobular carcinoma, where the members of the antioxidant protection pathway were changed [86]. Urine is also a valuable biofluid for metabolomics-based studies. In patients with malignant BC, increased levels of candidate metabolites have been reported in comparison to benign and healthy controls [238]. Metabolomics-based candidate biomarkers may be detected and quantified using gas chromatography-mass spectrometry (GC-MS) in the urine specimen [238]. For the early detection of BC, a study combining tissue transcriptomics, which emphasizes the gene expression profiles of cancer cells, with MS-based metabolomics targeting the metabolome profile of urine, highlighted its efficacy for BC biomarker identification [239].

Altered lipid metabolism impacts breast cancer cell growth and survival, plasticity, drug resistance, and metastasis, suggesting the potential use of lipidomics as a valuable diagnostic tool in BC [240]. As a part of the metabolomics field, lipidomics plays a key role in the discovery of the potential candidate biomarkers belonging to the several tumoral pathways invloved in breast cancer cell proliferation and survival [241]. Eiriksson et al., using a method of identifying the lipid species extracted from cultured cell lines via UPLC-QToF-MS, showed that BC subtypes defined by the transcriptome are reflected by the differences in the lipidome, emphasizing the potential of triacylglycerols for distinguishing between BC cell lines, as well as overexpressed phosphatidylcholine synthesis in TNBC cells [242]. Lipidomics-based analyses of the urinary phospholipids in patients with BC have been performed using the negative ion mode of nanoflow liquid chromatography-tandem mass spectrometry (nLC-ESI-MS/MS), suggesting that urinary lipids may serve in biomarker development [243]. Additionally, liquid chromatography-multiple reaction monitoring mass spectrometry (LC-MRM/MS) has identified significantly altered phospholipids in invasive ductal carcinoma (IDC) of the breast, suggesting a possible association with the invasive phenotype [241]. In situ desorption electrospray ionization-mass spectrometry imaging (DESI-MSI) lipidomic profiles of BC molecular subtypes and precursor lesions have shown that luminal B and TNBC subtypes emphasize more complex lipid profiles compared to luminal A and HER2 subtypes [244]. Moreover, DESI-MSI has identified a distinct lipid landscape between DCIS and invasive breast cancer (IBC) and across the molecular subtypes of BC [244].

### 2.5. Other Omics-Based Investigation of Breast Cancer

Dysregulated redox homeostasis is a hallmark of cancer [245]. Redoxomics is defined as a new omics focused on the study of the redoxome, which includes all the potentially oxidant chemical species, such as free radicals, reactive oxygen species (ROS), and reactive nitrogen species (NRS), as well as the antioxidant network that plays an essential role in all biological processes [246], including the regulation of proliferation and apoptosis [245]. BC patients emphasize an impaired oxidative/antioxidant state that facilitates oxidative stress (OS) [247], which plays a key role in the initiation, promotion, and progression of BC [248], in correlation with several biomarkers of DNA damage, lipid peroxidation, and protein damage [249]. Increased ROS levels have been found in TNBC cell lines, suggesting that ROS may serve as a potential target for therapy in the TNBC subtype [250]. However, ROS/RNS have a dual role in different stages of carcinogenesis [245].

Derived from proteomics, chaperomics investigates the chaperone system, as well as its functional partners and its involvement in carcinogenesis. Many molecular chaperones, such as HSP27, HSP60, HSP70, and HSP90, emphasize a protumorigenic role in BC and have become a valuable target for chaperonotherapy based on the inhibition of pro-cancer chaperones [251]. In relation to interactomics, in 2022, Ginsberg et al. introduced the term epichaperomics to define the link between stressor-induced protein interactome network modifications to the formation of the pathological scaffolds named epichaperomes in Alzheimer’s disease [4]. This new omics could also be used also for BC to define the functional changes in the interactome induced by stressors, suggesting that PPI changes may be drivers for disease-causing phenotypes. Immunotherapies have emphasized an essential role in BC therapy. Thus, tumor immunomics is focused on the integrated study of the tumor immune microenvironment (TIME), using immunogenomics, immunoproteomics, immune-bioinformatics, and other multi-omics data that reflect the immune landscape, especially based on single-cell-based technologies that enable an optimal molecular dissection of the TIME [252].

Microbiomics is focused on the characterization and quantification of the biomolecules responsible for the structure, function, and dynamics of microbial communities [253]. It is known that the breast, gut, and milk microbiomes are involved in the occurrence of malignant and non-cancerous breast lesions, suggesting that the microbiome is a valuable risk factor for BC [254], especially through the regulation of BC microenvironment [255]. Thus, Zhu et al. demonstrated that BC in postmenopausal women is associated with an altered gut metagenome [256], while modifications in the microbiota of human milk have consequences for mammary health [257].

BC has also been defined as an environmental disease [258] and ecological disorder [259], with eco-oncology improving our understanding of breast cancer biology [260]. Exposomics studies the exposome, the totality of human environmental exposure, complementing the genome [261,262]. Alcohol consumption [263], which stimulates the mobility of BC cells, the epithelial-mesenchymal transition (EMT), angiogenesis, OS and ROS [264], dioxin [265], instant coffee [266], ultra-processed foods [267], red meat [268], sugary drinks [269], hair dyes [270], endocrine disruptor chemicals [271], cigarette smoking [272,273], radiofrequency radiation [274], cellular phones [275], blue-light-radiation-emitting devices such as tablets and laptops [276], hormone replacement preparation [277] and hormone treatment [278], residential railway noise [279] and road traffic noise [280], house dust [281], viral infections [282], occupational exposure [283], polycyclic aromatic hydrocarbons (PAH) [284], and even a low water and liquid intake [285] have been positively and significantly associated with tumorigenesis and invasive BC risk. Ultraviolet radiation has also a role in the development of BC [286]. Foodomics is a comprehensive and high-throughput approach that exploits food science to improve human nutrition [287], with food being recognized as a powerful determinant in BC development [288]. Nutrigenomics integrates nutritional exoposomics and genomics, emphasizing the role of personalized diets in patients with a high risk of BC [289]. Flavonoids and other polyphenols act as epigenetic modifiers in BC [290], as well as the soy isoflavone genistein (GEN) that could reduce BC risk [291], the resveratrol, which is found in grapes and other food products [292] and prevents the epigenetic silencing of the *BRCA1* gene in human BC cells [293], and epigallocatechin gallate, a green tea major bioactive component that exerts a suppressive effect on BC [294]. Many xenobiotics promote the growth of human BC cells by inducing genetic modifications and epigenetic alterations, such as serum-derived estrogenic zeranol metabolites from industrial beef meat [295], as well as alcohol exposure [296]. Furthermore, a recent study highlighted an important correlation between fruit and vegetable consumption and the biomarkers of BC in lactating women, including weight, breast epithelial DNA methylation, and inflammation [297]. Taking into account that mammographic breast density is a recognized risk factor for BC, Pasta et al. showed that the association comprising boswellic acid, betaine, and myo-inositol significantly reduced mammary density, with relevant clinical outcomes in BC prevention [298].

Advances in genomics, transcriptomics, proteomics, and metabolomics have a great role in human reproduction and infertility treatment. Within the reproductomics field, fertility preservation and family planning in patients with BC represent the focuses in oncofertility, a modern branch of medicine with multidisciplinary characteristics [299,300].

## 3. Omics-Based Classification and Characterization of Breast Cancer Subtypes

The discovery and validation of novel noninvasive tools for the early diagnosis of BC would give the opportunity of subtype-specific targeted treatment [242]. The current clinical/routine using the classification of BC subtypes involves a semi-quantitative immunohistochemistry (IHC)-based analysis of hormone receptors, as well as other surface antigens, which cannot optimally address BC heterogeneity and drug resistance [301]. Transcriptomics-based techniques have also been extensively used for BC classification [233]. Gene expression does not generally reflect levels of proteins, so the identification of quantitative differences at the protein level has become more and more suitable for an accurate BC subtype classification. Today, an accurate classification of BC is essential for disease management and precise treatment. The discovery of appropriate tumor molecular classifiers significantly focuses on the application of omics approaches that analyze thousands of gene sequences, transcripts, or proteins in a single experiment [233]. However, the use of gene/transcript/protein expression profiling in routine clinical practice is not economical or practical, so many studies have focused on the use of panels of immunohistochemical biomarkers as surrogates or substitutes for the molecular classification of invasive breast cancers (IBC) [302]. Hence, the most commonly used IHC surrogates are cell surface protein hormone receptors (HR), such as estrogen receptor s(ER) and progesterone receptors (PR), as well as the human epidermal growth factor receptor 2 (ERBB2/HER2/NEU), dividing IBC into three major subtypes: ER+/PR+/ERBB2-, HER2/ERBB2-positive, and triple-negative/TNBC (ER-/PR-/ERBB2-) subtypes [302]. In 2000, Perou et al. proposed the first “molecular portrait of human breast tumors” based on complementary DNA (cDNA) microarrays that classified breast tumors into four molecular types [303]: luminal epithelial/estrogen receptor positive (ER+), ERBB2-overexpressing/HER2-enriched, basal epithelial-like, and normal breast-like. In 2001, Sørlie et al., using a highly similar cDNA-based approach, divided the previously characterized luminal epithelial/ER+ group into three main subgroups, luminal A, B, and C, each with a distinctive expression profile [304], leading to the characterization of six intrinsic subtypes of invasive BC, each unique in their incidence, survival, and response to therapy [302]. Thus, the molecular classification of BC subtypes emphasized a superior prognostic impact to traditional IHC [305]. In 2006, Zhiyuan et al., using Agilent oligo microarrays, created and validated a new BC intrinsic gene list (Intrinsic/UNC) that may be clinically used, which also showed overlap with previous breast tumor intrinsic gene sets, but also contained a proliferation signature that was not present in previous breast intrinsic gene sets [306]. In 2009, a 50-gene subtype predictor was developed using microarrays and quantitative reverse transcriptase polymerase chain reaction data (qRT-PCR), with the diagnoses by intrinsic subtype adding important prognostic and predictive information to the standard parameters for patients with BC [307]. Thus, BCs have been re-classified into five major intrinsic subtypes, luminal A (LumA), luminal B (LumB), HER2-enriched, basal-like, and normal-like, based on a 50-gene mRNA expression profile (PAM50 gene classification) [307]. In 2019, Mathews et al. proposed a classification of breast tumors into seven classes, defined by mRNA signatures: Basal/HER2, Basal/Myoepithelial (Myo), Myo/LumA, Myo/LumB, LumA, LumB/Basal, and Myo/LumB/HER2 [301,308]. Moreover, a unique miRNA-based-10 subtype taxonomy based on integrated data from protein, gene, and microRNA (miRNA) expressions has been proposed as the current gold standard to allow for the classification and separation of BC, emphasizing a strategy for BC therapy [301].

In 2020, Rohani and Eslahchi, using the deep embedded clustering (DEC) method, described four molecular subtypes of BC based on somatic mutation profiles: primary, progressive, proliferous, and perilous subtypes, each characterized by a specific gene signature [40]. Based on publicly available gene expression profiling data, six subgroups of ER+ breast tumors have been characterized for the improved understanding and treatment of ER+ BC [309]. Other genomics-based studies have analyzed plasma circulating tumor DNA (ctDNA) using NGS, emphasizing four subtypes of metastatic breast cancer (MBC): subtype 1, named extracellular function (ECF), is characterized by the aberrant genes involved in migration, invasion, angiogenesis, hematopoiesis, and immune regulation; subtype 2, or the cell proliferation (CP) subtype, groups the aberrant genes involved in apoptosis, cell cycle, metabolism, and development; subtype 3, named nucleus function (NF), involves the genes for DNA damage repair, epigenetics, RNA/protein assembly, and transcriptional regulation, while subtype 4, called the cascade signaling pathway (CSP), involves the aberrant genes involved in the hormone, PI3K/AKT, MAPK, JAK-STAT, and WNT pathways [310]. Genome-driven integrated classifications of BC, which integrate genomics- and transcriptomics-based analyses of BC to define ten integrative clusters associated with distinct clinical outcomes, have been published [38,311].

Genomics-based classifications of BC are widely used to identify intrinsic BC subtypes and recommend biomarkers for clinical use. However, newer proteomics-based classifications have been proposed to better reveal the functional phenotypic differences that lead to BC heterogeneity and provide accurate targeted therapies and clinical diagnostic tests [128]. Thus, highly sensitive proteomics-based techniques, such as SP3-CTP-Single-Pot, Solid-Phase-enhanced, Sample Preparation-Clinical Tissue Proteomics based on LC-MS/MS, which was applied to investigate 300 archival FFPE breast surgical specimens, revealed four distinct subtypes of TNBC: basal-immune hot, basal-immune cold, mesenchymal, and luminal with disparate survival outcomes [128]. Bouchal et al. used a highly multiplexed mode of targeted proteomics and a sequential windowed acquisition of all theoretical fragment ion spectra-mass spectrometries (SWATH-MS), known as a next-generation proteomics approach, to obtain digital proteome maps or “proteotypes” for a cohort of 96 breast tumor lysates, classifying them into five proteotype-based subtypes [233].

Metabolomics-based studies have revealed that BC cell subtypes present different metabolophenotypes, also called metabolic phenotypes, correlated with the current clinical classification [312]. Thus, three main metabolophenotypes have been described: the metabolophenotype 1, which emphasizes a glycolytic flux dependency specific for HR-positive cell lines (MCF7 and ZR751), the metabolophenotype 2, which develops a TCA cycle and mitochondrial oxidative metabolism dependency specific for TNBC cell lines (MDA-MB-231 and MDA-MB-468), and the metabolophenotype 3, which is specific for the HER2-positive cell line SKBR3 and emphasizes a mixed response [312]. Several contributions of different omics to BC classification and subtype characterization are listed in Table 3.

## 4. Omics-Based Applications in Breast Cancer Modeling

Breast cancer cell lines (BCCLs), despite having a known high-mutational frequency, are essential tools for studying cancer biology and heterogeneity [314]. They are widely used for BC modeling due to their molecular characteristic features, which remain almost the same with the corresponding subtype of primary breast tumors [315]. Integrated data from comprehensive genomics-transcriptomics-proteomics-based studies of genomic copy number variations (CNVs), mutations, mRNA expression, and protein expression have demonstrated that BCCLs emphasize some but not all the molecular features of breast primary tumors, adding more evidence for selecting BCCLs models that have the highest similarity with tumors for BC research [316]. Extensive omics datasets from human BCCLs, primary xenograft models of BC, primary BC specimens, and metastatic lesions have been developed [317]. Large-scale genomics data have emphasized surprising differences between BCCLs and metastatic BC tissue samples, for example in the case of the MDA-MB-231 cell line, which is widely recognized as a triple-negative, highly metastatic cell line that shows few genomic similarities to basal-like metastatic BC tumor samples, while other cell lines have been identified as closer models for different metastatic BC seen in the clinic [318]. A genomics-based study using the deep WGS of commonly used BCCLs and patient-derived xenografts (PDXs), also known as frequently used models in BC research, identified novel genomic alterations, such as point mutations and genomic rearrangements at base-pair resolution with biological significance [314]. Proteomics approaches based on the LC-MS/MS strategy have been used for the proteomic analyses of BCCL exosomes and have revealed disease patterns and potential biomarkers [201], as well as being used for examining BCCL-conditioned media that has shown a significant enrichment in the secreted proteins involved in BC development in comparison the corresponding cell lysates [319]. Several of our studies have investigated the effects of the overexpression [320] and downregulation [321,322] of the jumping translocation breakpoint (JTB) protein and its interacting partners for potential use as biomarker in BC, using an LC-MS/MS approach combined with in-gel [320,321], as well as the in-solution proteomics of MCF7 cells [322]. Three BCCLs, MCF10A (non-malignant), MCF7 (estrogen- and progesterone-receptor-positive, metastatic), and MDA-MB-231, have been investigated using LC-MS/MS for a phosphoproteomics-based analysis of BC-derived small EVs, in order to emphasize the disease-specific phosphorylated metabolic enzymes [323]. Applying affinity-purification MS to three BCCLs, Minkyu et al. delineated comprehensive biophysical interactomics networks for 40 frequently altered BC proteins, with and without relevant mutations. These resulting networks emphasized the cancer-specific protein–protein interactions (PPIs), interconnected and enriched for common and rare mutations, that were significantly reprogrammed by the introduction of key BC mutations [324]. A comparative analysis of PPI networks emphasized the differentially expressed genes and functional pathways that mediate BC metastasis to the brain and lung, using the MDA-MB-231 cell line derived from human tissues and its metastatic subpopulations, BrM2 and LM2 [325].

Panels of mRNAs, miRNAs, and protein biomarkers have been correlated with morphological differences for the identification of different BCCLs and the characterization of the luminal, HER2+, basal A and basal B, and TNBC cell line subtypes [315]. Moreover, 3D cell culture morphology of a panel of human BCCLs reflects the gene expression profile, as well as the protein expression patterns, in association with tumor cell invasiveness and the cell lines originating from metastases [326]. The single-cell transcriptomic profiling of BCCLs using single-cell RNA sequencing (scRNA seq) can be successfully used to capture the overall expressions of clinically relevant biomarkers, as well as to construct a BC atlas that may be used as reference to compare single-cell transcriptomics data from patients’ tissue biopsies and to perform BC subtype classifications and assessments of tumor intra-heterogeneity [327]. A miRomics-based study on the MCF7 and MDA-MB-231 BC cell lines using a qRT-PCR analysis showed that miR-139-5p has a significant role in BC cell motility and invasion, recommending it as a prognostic biomarker for aggressive forms of BC [328]. Metabolomics data based on the GC-MS profiling of metabolites report significant differences between BCCLs (MDA-MB-231, MDA-MB-453, and BT-474) and the MCF10A breast epithelial cell line, BC subtypes, and metabolic pathways such as amino acid metabolism, the TCA cycle, and glycolysis, thus showing that metabolic signatures have emerged as a promising biomarker and valuable tool for the understanding of the subtype-specific behavior of breast cancer [242,329,330,331]. Moreover, metabolomics emphasizes the effects of toxins, e.g., zearalenone, by their major metabolites (e.g., α-zearalenol) that have estrogenic proprieties, on BCCLs, in order to increase protein biosynthesis and lipid metabolism, as well as to induce ER^+^ BC progression [332]. UPLC-QToF-MS-based lipidomics of BCCLs have revealed differences between BC subtypes, demonstrating that the subtypes that are defined by specific transcriptomes are reflected in the differences in lipidome profiles [242]. In vitro explorations of the metabolic behavior of MCF7 cells (non-TNBC) and MDA-MB-231 (TNBC) cells under lipidomic based LC-MS have shown significant differences in lipid regulation, which may be associated with the aggressiveness and difficulties of treating TNBC, emphasizing phosphatidylethanolamine as biomarker of TNBC [331].

## 5. Omics-Based Investigations of the Tumoral Suppressor *TP53*

First, cancer research is focused on identifying carcinogenic mutations and emphasizing how they are involved in tumor progression [324]. *TP53* is the most frequently mutated gene in human cancer [332], including BC [333]. The *TP53* gene is located on the human chromosome 17 at p13 and encodes the transcription factor *TP53*/p53, also known as tumor protein 53, cellular tumor antigen p53, or transformation-related protein 53. *TP53* is a key tumor suppressor that is inactivated in almost all cancers due to the missense mutations in the *TP53* gene or an overexpression of its negative regulators [334]. *TP53* is essential for protecting the genome from alterations and instability, which encourages tumorigenesis [335]. *TP53* mutations result in structural p53 protein destabilization, causing its partial unfolding and deactivation or an impairment of its DNA-binding proprieties [334]. A very high risk of BC has been reported in women that carry germline mutations in their *TP53* gene [336]. Thus, mutations in p53, such as exon 4 and intron 3, which are reported as being frequently mutated in BC patients [333], are associated with advanced stages or more malignant BC subtypes, such as TNBC [337]. Moreover, mutations in p53 have been associated with endocrine therapy resistance, poor prognoses, and mutations in stress kinase MAP3K1 and GATA3, a zinc finger transcription factor [44]. In addition to mutated p53, the PI3K/AKT pathway is also deregulated in the majority of TNBC, which causes the over-activation of AKT, which leads to cancer development [337]. A Multi-Patient-Targeted (MPT) single-cell DNA sequencing (scDNA-seq) approach was developed for the identification of early *TP53* mutations in TNBC tissue samples [110]. p53 protein facilitates DNA repair, cell cycle arrest, or apoptosis following DNA damage [334], orchestrating very diverse cellular responses to different types of stress [338]. According to uniprot.org, the molecular functions of this protein are diverse: DNA binding/histone binding and transcription regulator activity. *TP53* is involved in various biological processes, such as the immune system process, DNA repair, DNA recombination, the regulation of DNA-template transcription, cell signaling, anatomical structure development, and protein-containing complex assembly. It is located in the nucleus. *TP53*’s capacity to respond to different stress-related stimuli depends on the expression of its isoforms [338]. Some smaller p53 isoforms inhibit the wild-type p53 [339], so the unbalancing expressions of different p53 splice variants are involved in tumorigenesis [332] and, moreover, are related to the clinical features of BC and its outcomes [339,340]. In the aggressive TNBC subtype, the p53/40 isoform was reported as being significantly overexpressed in the tumor tissue compared to the normal one, while the expression of the p53/p47 isoform, an alternative translation initiation variant, was induced during the unfolded protein response following the endoplasmic reticulum stress [338]. In BC cells with mutated p53, a specific inhibitor of histone deacetylase 6, ACY-1215, caused the acetylation of p53 [337]. Phosphorylation and acetylation activate p53. Activated p53 stimulates the transcription of a variety of downstream genes, leading to cell growth, arrest, DNA repair, and apoptosis. *TP53* interactomics has shown that its mutations lead to altered expressions of various genes that are under the transcriptional control of this gene [333].

## 6. Conclusions

The main hallmarks of cancer are very complex: genomic instability and mutation, the sustaining of proliferative signaling, evading growth suppressors, resisting cell death, enabling replicative immortality, inducing or accessing vasculature, activating invasion and metastasis, deregulated cellular metabolism, avoiding immune destruction, tumor-promoting inflammation, unlocking phenotypic plasticity, non-mutational epigenetic reprogramming, polymorphic microbiomes, and senescent cells [338]. Omics-based investigations, either single omics or large-scale multi-omics, have been expanding each year [341], allowing for in-depth investigations of the molecular bases of BC phenotypes, BC hallmarks, carcinogenesis, progression, and metastasis, which are necessary for accurate diagnoses, prognoses, and therapy assessments in predictive, precision, and personalized oncology. This review discussed both traditional and several new omics fields involved in modern BC investigations, which may all be integrated as a holistic term, onco-breastomics.

Central dogma-based omics fields, such as genomics, transcriptomics, and proteomics have exploited genome-wide association studies (GWAS), transcriptome-wide association studies (TWAS), and proteome-wide association studies (PWAS) for an in-depth understanding of BC hallmarks, using the rapid and recent advances in molecular profiling strategies and analytical techniques based on high-throughput sequencing and mass spectrometry (MS). Non-mutational epigenetic reprogramming studies have led to the emergence of three epiomics-based multidimensional approaches that provide opportunities for gaining insights into BC heterogeneity and its underlying mechanisms. The main epiomics fields (epigenomics, epitranscriptomics, and epiproteomics) have dechiphered the eipgenetic DNA changes, detected RNA modifications, and investigated the posttranslational modifications (PTMs) affecting protein functions for an in-depth understanding of cancer cell proliferation, migration, and invasion. Novel omics fields, such as epichaperomics or epimetabolomics, could investigate the modifications in the interactome induced by stressors and provide PPI changes or the metabolites removed from their classical function as drivers of BC-causing phenotypes. An investigation of polymorphic microbiomes led to the development of metagenomics, which showed that the microbial communities of the human body may serve as potential biomarkers for prognoses and diagnoses or help with the development of new therapeutic strategies. Over the last years, several proteomics-derived omics, such as matrisomics, exosomics, secretomics, kinomics, and phosphoproteomics, have provided valuable data for a deeper understanding of the dysregulated pathways in BC cells and their tumor microenvironment (TME). Several hyphenated omics approaches, such as proteo-genomics, proteo-transcriptomics, or phosphoproteomics-exosomics, are useful for the identification of putative BC biomarkers and therapeutic targets. Today, advanced technologies facilitate the exploration of BC heterogeneity through multiple single-cell omics into a single experiment that can overcome the insufficient data that has resulted from one single omics approach. Thus, scRNA-seq enables the spatial mapping of BC heterogeneity and the novel cellular interactions in the BC microenvironment, which enables breast cancer stratification into various “ecotypes” with specific cellular compositions and clinical outcomes. To develop non-invasive diagnostic tests and discover new biomarkers for BC, classic omics-based strategies have allowed for significant advances in blood/plasma-based omics. Salivaomics, urinomics, and milkomics have appeared as integrative omics that may develop a high potential for the early detection of and home-testing strategies for BC, especially based on non-invasive proteomic biomarkers. Thus, analyses of the tumor circulome are consideredas novel frontiers in liquid biopsies. Last but not least, interactomics promotes a holistic view in BC research via the construction and characterization of the protein–protein interaction (PPI) networks that provide a novel hypothesis for the pathophysiological processes involved in BC progression and subtyping. Omics-based investigations have applications in BC modeling, as well as accurate BC classification and subtype characterization. The future in omics-based investigations of BC may also be focused on multi-omics single-cell analyses.

## Figures and Tables

**Table 2 molecules-28-04768-t002:** Proteomics-based and proteomics-derived studies in breast cancer.

Proteomics-Based and Proteomics-Derived Investigation of BC	Samples	Omics-Based Techniques	StudiesRelevance	References
**proteomics** 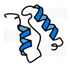	FFPE	SP3-CTP;LC-MS/MS	high sensitive MS-based methodology for capturing biological features in FFPE tumor samples; characterization of BC heterogeneity in a clinically-applicable manner, biomarkers and therapeutic targets discovery, clinical BC classification	[128]
FF	SWATH-MS(LC-MS/MS)	highly multiplexed mode of targeted proteomics that generated large-scale quantitative proteomics profiles of BC tissues; BC classification into proteotype-based subtypes with different treatment strategies	[233]
blood/serum/plasma	LC-ESI-MS/MS	comparison between peptides and proteins specific to BC plasma and ovarian cancer and matched controls	[70]
tumor interstitial fluid	LC-MS/MS	high-throughput proteomics for identification of tumor subtype-specific relevant biomarkers	[75]
saliva and serum samples	iTRAQLC-ToF-MS/MS	identification of protein biomarkers for early detection of BC; platform for investigating the responsive proteomic profile in benign and malignant breast tissue using saliva and serum from the same women	[227]
urine	label free LC-MS/MS	identification of protein biomarkers for early screening detection and monitoring invasive BC progression	[90]
colostrum and milk	nLC-MS/MS	BC biomarkers discovery	[234]
NAF;NAF spots on Guthrie cards	SELDI-ToF-MS;1D-LC-MS/MS	identification of differential proteomic profile between women with/without BC; BC biomarkers identification; identification of NAF proteome associated with BC development	[125,126,235]
**salivaomics: transcriptomics and proteomics**	saliva of BC patients vs. matched controls	proteomics: 2D-DIGE,MALDI-ToF MS;transcriptomics: Affymetrix HG-U133-Plus-2.0 Array, RT-qPCR	mRNA biomarkers and one protein biomarker were pre-validated on the preclinical validation sample set for BC detection	[123]
**phosphoproteomics** 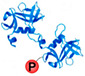	FF	Fe-IMAC,iTRAQSCX LC-ESI-MS/MS SID-SRM-MS for validation	large-scale phosphoproteome quantification in high- and low-risk recurrence groups as powerful tool for biomarker discovery using clinical samples	[215]
FFPE,TNBC cell lines,mouse models (PDXs)	nano-LC-MS/MS	high-throughput phosphoproteomics for target-based clinical classification system for TNBC	[213]
**kinomics, phosphoproteomics, proteomics, transcriptomics** 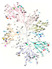	PDX models of TNBC	RPPA,LC-MS/MS;MS-based kinome profiling	integrative phosphoproteogenomic analysis for identification of intrinsic resistance mechanisms of TNBC to PI3K inhibition	[217]
**exosomics** 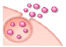	plasma and total blood	MALDI-ToF/ToF MS	proteomic analysis of exosomes for BC diagnostic/prognostic biomarkers or novel therapeutic targets	[203]
breast cell line derived exosomes	nanoLC-MS/MS	proteomic profile of cancerous and non-tumorigenic breast cell lines for BC diagnostic/prognostic biomarker discovery	[201]
**secretomics, matrisomics**	human breast samples (normal and IDC)	LC-SRM,LC-MS/MS,TPM, SHG, two-photon fluorescence imaging	targeted matrisome analysis for compositional change in matrisome proteins according to collagen re-organization during BC progression; candidate proteins involved in collagen alignment	[197]
LC-MS/MS,MALDI-FT-ICR MS,MALDI-ToF MS,MALDI-MS/MS	proteomic remodeling of TME; review of significant dysregulated proteins involved in TME remodelling in IDC	[196]
**phosphoproteomics and exosomics**	plasma samples	LC-MS/MS	phosphoproteomic profile of EVs of patients and healthy controls for potential biomarkers to differentiate BC patients from healthy controls	[27]
**interactomics** 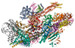	serum and saliva	network biology approach	PPI networks for proteins in serum and saliva for potential biomarkers in BC diagnosis and prognosis	[227]

Abbreviations: BC—breast cancer; Fe—IMAC-immobilized Fe (III) affinity chromatography; FF—fresh frozen; FFPE—formalin-fixed paraffin-embedded; IDC—invasive ductal carcinoma; iTRAQ—isobaric tag for relative quantification; LC-ESI-MS/MS—liquid chromatography electrospray ionisation tandem mass spectrometry; LC-SRM—liquid chromatography-selected reaction monitoring; NAF—nipple aspirate fluid; PDX—patient derived xenograft; RPPA—reverse-phase protein array; SCX—strong cation exchange chromatography; SID-SRM—stable isotope dilution-selected reaction monitoring; SHG—second-harmonic generation; SP3-CTP—Single-Pot, Solid-Phase-enhanced, Sample Preparation-Clinical Tissue Proteomics; SWATH-MS—sequential windowed acquisition of all theoretical fragment ion spectra-mass spectrometry; TNBC—triple-negative breast cancer; TME—tumor microenvironment; and TPM—two-photon microscopy.

**Table 3 molecules-28-04768-t003:** Contribution of different omics to BC classification and subtypes characterization.

Omics	Year of Publication	Samples	Techniques	BC Subtypes	Studies Relevance	References
**Transcriptomics**	2000	surgical specimens and cultured cell lines	cDNA microarrays	basal epithelial-like, ERBB2-overexpressing, normal breast-like, luminal epithelial/ER+	“molecular portrait of human breast tumors”	[303]
	2001	FF tissue samples		basal epithelial-like, ERBB2-overexpressing, normal breast-like, luminal A, luminal B, luminal C	“breast tumor intrinsic” subtypes classification; poor prognosis for basal-like subtype, and significant difference in outcome for two ER+ groups	[304]
	2006	FF breast tumor samples	Agilent oligo microarrays	LumA, LumB, basal-like, HER2+/ER−, normal breast-like	validation of “breast tumor intrinsic” subtype classification	[306]
	2009	FFPE, FF	qRT-PCR, microarray	LumA, LumB, HER2-enriched, basal-like, normal-like	BC intrinsic molecular subtypes defined by mRNA expression of 50 genes (PAM50 risk assessment tool)	[307]
**miRomics**	2021	TCGA, METABRIC, PAM50 mRNA, GTEx datasets		Basal, Basal-HER2, Basal-LumB, Basal-LumA, HER2, HER2-LumB, HER2-LumA, LumA-LumB, LumA, LumB	categorization of breast tumor samples based on miRNA expression profiling	[301]
**Genomics**	2020	861 breast tumors	cancer genome atlas (TCGA) database	primary, progressive proliferous perilous	discovery of the molecular subtypes of BC using somatic mutation profiles of tumors	[40]
	2022	223 patients with MBC	NGS for ctDNA	**subtype 1**: extracellular function (ECF), **subtype 2**: cell proliferation (CP), **subtype 3**: nucleus function (NF), **subtype 4**: cascade signaling pathway (CSP)	HR/HER2 subtyping of MBC based on 70 plasma ctDNA alterations	[310]
**Genomics and transcriptomics**	2012, 2013	2000 breast tumors	germline variants (CNVs and SNPs) and somatic aberrations (CNSAs) associated with alteration in gene expression	10 novel molecular subgroups	novel molecular classification of the BC population based on the impact of somatic CNAs on the transcriptome	[38,311]
**Proteomics and transcriptomics**	2019	FF tissue samples	SWATH-MS (LC-MS/MS)	five proteotypes-based BC subtypes	SWATH proteotype pattern largely recapitulate the conventional BC subtypes; TNBC are most heterogeneous in protein expression	[233]
**Proteomics**	2022	archival FFPE tumor samples	SP3-CTP-MS (LC-MS/MS)	**BL-BC subtypes**: basal-immune hot and basal-immune cold; **TNBC subtypes**: basal-immune hot, basal-immune cold, mesenchymal, and luminal; **HER2-enriched** groups differing by ECM, lipid metabolism, and immune-response	potential biomarkers for existing chemotherapies or emerging immunotherapies	[128]
**Metabolomics**	2021	BC cell lines	LC-MS	three BC metabolophenotypes (1, 2, and 3): **metabolophenotype 1**: glycolytic flux dependency specific for HR-positive cell lines (MCF7 and ZR751); **metabolophenotype 2**: TCA cycle and mitochondrial oxidative metabolism dependency specific for TNBC cell lines (MDA-MB-231 and MDA-MB-468); **metabolophenotype 3**: specific for HER2-positive cell line SKBR3 with mixed response	BC cell types display different metabolophenotypes correlated with the current clinical classifications	[312]
**Metabolomics and transcriptomics**	2010	BC tissue samples (IDC, ER+, luminal A)	HR MAS MRS, gene expression microarrays	three types of luminal A BC (A1, A2, and A3); **A2 subgroup**, a more aggressive BC: higher glycolytic activity/higher Warburg effect, cell cycle, and DNA repair	transcriptional and metabolic subtyping based on high-dimensional data	[121]
**Metabolomics, genomics, and proteomics**	2016	primary breast carcinoma FF samples	HR MAS MRS, RPPA, mRNA expression profiling, integrated pathway analysis	three metabolic clusters (Mc1, Mc2, and Mc3); **Mc1**: highest levels of GPC and PCho, downregulation of genes related to collagens and ECM; **Mc2**: highest levels of glucose, overexpression of genes related to collagens and ECM; **Mc3**: highest levels of lactate and alanine, overexpression of genes related to collagens and ECM	information about the heterogeneity of BCs, susceptibility to different metabolically targeted drugs	[313]
**Salivaomics**	2022	saliva	biochemical analysis/ biochemical indicators	BL-BC was defined of the maximum number of indicators; HER2+/HER2- and ER+ BC differ from the control group; ER/PR+ BC group has more favorable ratio of biochemical indicators compared to ER/PR—BC	12 biochemical indicators	[28]

Abbreviations: BC—breast cancer; BL-BC—basal-like BC; cDNA—complementary/copy DNA; CNAs—copy number aberrations; CNVs—copy number variants; ctDNA—circulating tumor DNA; ECM—extracellular matrix; GPC—glycerophosphocholine; HR/HER2—hormone receptor/human epidermal growth factor receptor 2; HR MAS MRS—high-resolution magic-angle spinning magnetic resonance spectroscopy; IDC—invasive ductal carcinoma; LC-MS/MS—liquid chromatography tandem mass spectrometry; METABRIC—Molecular Taxonomy of Breast Cancer International Consortium; MBC—metastatic BC; NGS—next generation sequencing; PAM50—Prediction Analysis of Microarray; PCho—phosphocholine; qRT-PCR—quantitative reverse transcriptase polymerase chain reaction; RPPA—reverse phase protein array; SNP—single-nucleotide polymorphism; SP3-CTP—Single-Pot, Solid-Phase-enhanced, Sample Preparation-Clinical Tissue Proteomics; SSP—single sample predictions; SWATH-MS—sequential windowed acquisition of all theoretical fragment ion spectra-mass spectrometry; TCA—tricarboxylic acid cycle; TGCA—The Cancer Genome Atlas; and TNBC—triple-negative breast cancer.

## Data Availability

Not applicable.

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
