# Peer review of "Omics-Based Investigations of Breast Cancer"

_molecules, 2023, doi:10.3390/molecules28124768_

Round 1

Reviewer 1 Report

Dear authors,

I read with great interest the manuscript, which falls within the aim of this Journal. In my honest opinion, the topic is interesting enough to attract the readers’ attention. Nevertheless, authors should clarify some points and improve the discussion, as suggested below.

In my opinion you have to improve the paper refering in the text how usually these pts need to preserve their fertility by freezing their oocytes for future family planning.

I also suggest to refer how in case of early breast density the literaure report how a nutraceutical supplemetation can be helpfull to reduce it.

I suggest to read and cite these pertinents articles:

Oncofertility and Reproductive Counseling in Patients with Breast Cancer: A Retrospective Study

An association of boswellia, betaine and myo-inositol (Eumastós®) in the treatment of mammographic breast density: A randomized, double-blind study

Update on fertility preservation for younger women with breast cancer

Fertility preservation in female cancer sufferers: (only) a moral obligation?

Authors should consider the following recommendations:

I suggest you to improve the paper refering to the updated literature and how

Minor editing of English language required

Author Response

Answers reviewer 1

I read with great interest the manuscript, which falls within the aim of this Journal. In my honest opinion, the topic is interesting enough to attract the readers’ attention. Nevertheless, authors should clarify some points and improve the discussion, as suggested below.

Answer: thank you for taking the time to read our manuscript and for giving us comments that improved the quality of our manuscript. We addressed the reviewer’s comments and the answers are outlined below:

In my opinion you have to improve the paper refering in the text how usually these pts need to preserve their fertility by freezing their oocytes for future family planning. I also suggest to refer how in case of early breast density the literaure report how a nutraceutical supplemetation can be helpfull to reduce it.

Answer: we believe that this was a great suggestion for improving the quality of our manuscript. Among a lot of references that demonstrate the role of exposomics in BC risk (subchapter 2.5), we included many papers, in which we discuss the effect of maternal diet affects not only the moms’ breast health, but also the quality of milk and the wellbeing of infants.

“BC was also defined as an environmental disease [1] and ecological disorder [2], the eco-oncology improving the understanding breast cancer biology [3]. Exposomics studies the exposome, the totality of human environmental exposure, complementing the genome [4], [5]. Alcohol consumption [6], which stimulates the mobility of BC cells, the epithelial-mesenchymal transition (EMT), angiogenesis, OS and ROS [7], dioxin [8], instant coffee [9], ultra-processed foods [10], red meat [11], sugary drinks [12], hair dyes [13], endocrine disruptor chemicals [14], cigarette smoking [15], [16], radiofrequency radiation [17], cellular phones [18] and blue light radiation emitting devices, such as tablet and laptop [19], hormone replacement preparation [20] and hormone treatment [21], residential railway noise [22] and road traffic noise [23], house dust [24], viral infections [25], occupational exposure [26], polycyclic aromatic hydrocarbons (PAH) [27] and even a low water and liquids intake [28] were positively and significantly associated with tumorigenesis and invasive BC risk. Ultraviolet radiation has also a role in the development of BC [29]. Foodomics is a comprehensive and high-throughput approach that exploits food science to improve human nutrition [30], food being recognized as a powerful determinant in BC development [31]. Nutrigenomics integrates nutritional exoposomics and genomics, emphasizing the role of personalized diets in patients with high risk of BC [32]. Flavonoids and other polyphenols act as epigenetic modifiers in BC [33], as well as the soy isoflavone genistein (GEN) that could reduce the BC risk [34], the resveratrol, found in grapes and other food products [35], that prevents the epigenetic silencing of BRCA1 gene in human BC cells [36], and epigallocatechin gallate, a green tea major bioactive component that exerts a suppressive effect on BC [37]. Many xenobiotics promote the growth of human BC cells by inducing genetic modifications and epigenetic alterations, such as serum-derived estrogenic zeranol metabolites from industrial beef meat [38], as well as the alcohol exposure [39]. Also, a recent study highlighted an important correlation between fruit and vegetable consumption and biomarkers of BC in lactating women, including weight, breast epithelial DNA methylation and inflammation [40]. Taking account that the mammographic breast density is a recognized risk factor for BC, Pasta et al. showed that the association comprising boswellic acid, betaine, and myo-inositol significantly reduced the mammary density with relevant clinical outcomes in BC prevention [41]. Advances in genomics, transcriptomics, proteomics and metabolomics have a great role in human reproduction and infertility treatment. Within the reproductomics field, fertility preservation and family planning in patients with BC represent the focuses in oncofertility, a modern branch of medicine with multidisciplinary characteristics [42, 43].”

I suggest to read and cite these pertinents articles:

-Oncofertility and Reproductive Counseling in Patients with Breast Cancer: A Retrospective Study

-An association of boswellia, betaine and myo-inositol (Eumastós®) in the treatment of mammographic breast density: A randomized, double-blind study

Update on fertility preservation for younger women with breast cancer

-Fertility preservation in female cancer sufferers: (only) a moral obligation?

Answer: we discussed these three studies, including a discussion about exposomics and reproductomics within the subchapter 2.5. (Other omics-based investigation of breast cancer) : “Taking account that the mammographic breast density is a recognized risk factor for BC, Pasta et al. showed that the association comprising boswellic acid, betaine, and myo-inositol significantly reduced the mammary density with relevant clinical outcomes in BC prevention [41]. Advances in genomics, transcriptomics, proteomics and metabolomics have a great role in human reproduction and infertility treatment. Within the reproductomics field, fertility preservation and family planning in patients with BC represent the focuses in oncofertility, a modern branch of medicine with multidisciplinary characteristics [42, 43].

Authors should consider the following recommendations:

I suggest you to improve the paper refering to the updated literature.

Answer: we improved the quality by updating it with the newer literature, including the reviewer’s suggestion.

References

  1. Lynn, H., et al., Breast Cancer: an environmental disease. The case for primary prevention2005.
  2. Hiatt, R.A. and J.G. Brody, Environmental Determinants of Breast Cancer. Annual Review of Public Health, 2018. 39(1): p. 113-133.
  3. Reynolds, B.A., M.W. Oli, and M.K. Oli, Eco-oncology: Applying ecological principles to understand and manage cancer. Ecology and Evolution, 2020. 10(16): p. 8538-8553.
  4. Vrijheid, M., The exposome: a new paradigm to study the impact of environment on health. Thorax, 2014. 69(9): p. 876-878.
  5. Bessonneau, V. and R.A. Rudel, Mapping the Human Exposome to Uncover the Causes of Breast Cancer. International journal of environmental research and public health, 2019. 17(1): p. 189.
  6. McDonald, J.A., A. Goyal, and M.B. Terry, Alcohol Intake and Breast Cancer Risk: Weighing the Overall Evidence. Current breast cancer reports, 2013. 5(3): p. 10.1007/s12609-013-0114-z.
  7. Wang, Y., et al., Cellular and Molecular Mechanism Underlying Alcohol-induced Aggressiveness of Breast Cancer. Pharmacol Res. 2017 Jan;115:299-308, 2017. 115.
  8. Vopham, T., et al., Dioxin exposure and breast cancer risk in a prospective cohort study. Environmental Research, 2020. 186: p. 109516.
  9. Lee, P.M.Y., et al., Associations between Coffee Products and Breast Cancer Risk: a Case-Control study in Hong Kong Chinese Women. Scientific Reports, 2019. 9(1): p. 12684.
  10. Fiolet, T., et al., Consumption of ultra-processed foods and cancer risk: results from NutriNet-Santé prospective cohort. BMJ (Clinical research ed.), 2018. 360: p. k322-k322.
  11. Lo, J., et al., Association between meat consumption and risk of breast cancer: Findings from the Sister Study. International Journal of Cancer, 2019. 146.
  12. Chazelas, E., et al., Sugary drink consumption and risk of cancer: results from NutriNet-Santé prospective cohort. BMJ (Clinical research ed.), 2019. 366: p. l2408.
  13. GERA, R., et al., Does the Use of Hair Dyes Increase the Risk of Developing Breast Cancer? A Meta-analysis and Review of the Literature. Anticancer Research, 2018. 38(2): p. 707-716.
  14. Eve, L., et al., Exposure to Endocrine Disrupting Chemicals and Risk of Breast Cancer. International Journal of Molecular Sciences, 2020. 21(23): p. 9139.
  15. Jones, M.E., et al., Smoking and risk of breast cancer in the Generations Study cohort. Breast Cancer Research, 2017. 19(1): p. 118.
  16. Huynh, D., et al., Electronic cigarettes promotes the lung colonization of human breast cancer in NOD-SCID-Gamma mice. International journal of clinical and experimental pathology, 2020. 13(8): p. 2075-2081.
  17. Shih, Y.-W., et al., Exposure to radiofrequency radiation increases the risk of breast cancer: A systematic review and meta-analysis. Experimental and therapeutic medicine, 2021. 21(1): p. 23-23.
  18. West, J.G., et al., Multifocal Breast Cancer in Young Women with Prolonged Contact between Their Breasts and Their Cellular Phones. Case reports in medicine, 2013. 2013: p. 354682-354682.
  19. Mortazavi, A.R. and S.M.J. Mortazavi, Women with hereditary breast cancer predispositions should avoid using their smartphones, tablets and laptops at night. IJBMS, 2018. 21: p. 112-115.
  20. Vinogradova, Y., C. Coupland, and J. Hippisley-Cox, Use of hormone replacement therapy and risk of breast cancer: nested case-control studies using the QResearch and CPRD databases. BMJ (Clinical research ed.), 2020. 371: p. m3873.
  21. de Blok, C.J.M., et al., Breast cancer risk in transgender people receiving hormone treatment: nationwide cohort study in the Netherlands. BMJ (Clinical research ed.), 2019. 365: p. l1652.
  22. Sørensen, M., et al., Road and railway noise and risk for breast cancer: A nationwide study covering Denmark. Environmental Research, 2021. 195: p. 110739.
  23. Andersen, Z., et al., Long-term exposure to road traffic noise and incidence of breast cancer: a cohort study. Breast Cancer Research, 2018. 20.
  24. Xiang, P., et al., Organic extract of indoor dust induces estrogen-like effects in human breast cancer cells. Science of The Total Environment, 2020. 726: p. 138505.
  25. Gannon, O.M., et al., Viral infections and breast cancer – A current perspective. Cancer letters, 2018. 420: p. 182-189.
  26. Ekenga, C.C., et al., Breast Cancer Risk after Occupational Solvent Exposure: the Influence of Timing and Setting. Cancer research, 2014. 74(11): p. 3076-3083.
  27. Shen, J., et al., Dependence of cancer risk from environmental exposures on underlying genetic susceptibility: an illustration with polycyclic aromatic hydrocarbons and breast cancer. British journal of cancer, 2017. 116(9): p. 1229-1233.
  28. Keren, Y., et al., Investigation of the Association between Drinking Water Habits and the Occurrence of Women Breast Cancer. International journal of environmental research and public health, 2020. 17(20): p. 7692.
  29. Hiller, T.W.R., et al., Solar Ultraviolet Radiation and Breast Cancer Risk: A Systematic Review and Meta-Analysis. Environmental health perspectives, 2020. 128(1): p. 016002.
  30. Capozzi, F. and A. Bordoni, Foodomics: a new comprehensive approach to food and nutrition. Genes & Nutrition, 2013. 8(1): p. 1-4.
  31. Regal, P., et al., Food and omics: unraveling the role of food in breast cancer development. Current Opinion in Food Science, 2021. 39: p. 197-207.
  32. Sellami, M. and N.L. Bragazzi, Nutrigenomics and Breast Cancer: State-of-Art, Future Perspectives and Insights for Prevention. Nutrients, 2020. 12(2): p. 512.
  33. Selvakumar, P., et al., Flavonoids and Other Polyphenols Act as Epigenetic Modifiers in Breast Cancer. Nutrients, 2020. 12(3): p. 761.
  34. Rahal, O.M. and R.C.M. Simmen, PTEN and p53 cross-regulation induced by soy isoflavone genistein promotes mammary epithelial cell cycle arrest and lobuloalveolar differentiation. Carcinogenesis, 2010. 31(8): p. 1491-1500.
  35. Fustier, P., et al., Resveratrol increases BRCA1 and BRCA2 mRNA expression in breast tumour cell lines. British journal of cancer, 2003. 89(1): p. 168-172.
  36. Papoutsis, A.J., et al., Resveratrol prevents epigenetic silencing of BRCA-1 by the aromatic hydrocarbon receptor in human breast cancer cells. The Journal of nutrition, 2010. 140(9): p. 1607-1614.
  37. Huang, Y.-J., et al., Protective Effects of Epigallocatechin Gallate (EGCG) on Endometrial, Breast, and Ovarian Cancers. Biomolecules, 2020. 10(11): p. 1481.
  38. ZHONG, S., et al., Serum Derived from Zeranol-implanted ACI Rats Promotes the Growth of Human Breast Cancer Cells <em>In Vitro</em>. Anticancer Research, 2011. 31(2): p. 481-486.
  39. Wang, J., et al., Alcohol consumption and breast tumor gene expression. Breast Cancer Research, 2017. 19(1): p. 108.
  40. Sturgeon, S., et al., New Moms Wellness Study: the randomized controlled trial study protocol for an intervention study to increase fruit and vegetable intake and lower breast cancer risk through weekly counseling and supplemental fruit and vegetable box delivery in breastfeeding women. BMC Women's Health, 2022. 22.
  41. Gullo, G., et al., An association of boswellia, betaine and myo-inositol (Eumastós®) in the treatment of mammographic breast density: A randomized, double-blind study. European review for medical and pharmacological sciences, 2015. Vol. 19 -: p. 4419-4426.
  42. Zaami, S., et al., Oncofertility and Reproductive Counseling in Patients with Breast Cancer: A Retrospective Study. Journal of clinical medicine, 2022. 11(5): p. 1311.
  43. Zaami, S., et al., Fertility preservation in female cancer sufferers: (only) a moral obligation? The European Journal of Contraception & Reproductive Health Care, 2022. 27(4): p. 335-340.

Reviewer 2 Report

Anca-Narcisa Neagu and colleagues recently published an intriguing review on breast cancer research utilizing omics technology. This comprehensive review effectively highlights the recent advancements in the field of breast cancer, specifically by providing an extensive list of potential omics methods. However, it would be beneficial for the breast cancer community if the review delved further into discussing how we can utilize the data, information, and knowledge obtained through these omics to facilitate future studies, rather than solely presenting a list of methods.

I have a few questions in light of this review:

Are there any noteworthy examples demonstrating the clinical benefits of utilizing omics-based methods, beyond classification?

What are the pros and cons researchers should consider when selecting an omics method, such as sample size, sensitivity, cost, and stages of breast cancer?

Which omics analysis tools/platforms/websites have the potential to provide more comprehensive information?

Additionally, considering the growing body of evidence suggesting the involvement of the microbiota in breast cancer progression and metastasis, it would be valuable to include a discussion on microbiomics in this review.

Extensive editing of English language required

Author Response

Answers reviewer 2

Are there any noteworthy examples demonstrating the clinical benefits of utilizing omics-based methods, beyond classification?

Answer: We included in our manuscript the following explanations: Untargeted quantitative proteomics is the first step in identification of the dysregulated proteins in two different conditions. However, once this is done and we have the list of the dysregulated proteins, then they are potential biomarkers. Verification of these biomarkers can then be done using targeted quantitative proteomics.

Targeted proteomics looks for specific peptide sequences to quantify the amount of protein within a sample, and between disease and healthy states. Single reaction monitoring (SRM) and multi reaction monitoring (MRM) are two ways to quantify protein levels between two samples. Single reaction monitoring is only looking for one peptide sequence or only one m/z, whereas MRMs are looking to quantify multiple peptide sequences, therefore scanning over many m/z ratios based on the method development. This is just one way to utilize proteomic methods to quantify protein level in a sample.

Each omics-based method will have different ways in which there are clinical benefits beyond classification. Omics-based methods are able to do much more than just classification. There are discovery experiments in proteomics, such as biomarkers, but one can also look at the metabolome, protein post-translational modifications, or protein-protein interactions to determine what is happening in the body. One can take metabolites and track that back to a specific drug or process that is happening within the body. All omics-based methods are able to do much more than just classification.

Another relatively new method is to investigate pharmacokinetics and pharmacodynamics of protein-based drugs in the blood stream and at the tumor level (tumor microenvironment). Administering a protein drug (i.e., antibodies or protein therapeutics) requires its quantification in blood over the duration of the treatment, which it is done by targeted quantitative proteomics, discussed earlier (SRM and MRM). There is no better and more precise method than targeted proteomics. The only alternative option is ELISA, which can be specific if the antibodies are good, but we know that most of the antibodies on the marker can be less specific as they should be, or even unspecific. So, yes, targeted mass spectrometry-based quantitative proteomics or metabolomics are the best options for the future, with multiomics to follow.    

What are the pros and cons researchers should consider when selecting an omics method, such as sample size, sensitivity, cost, and stages of breast cancer?

Answer: We included in our manuscript the following explanations: When choosing an omics-based method, we first need to think of the outcome needed. Are we looking for determining the proteome, metabolome or genome etc. Each omics method has differing amounts of time from sample preparation to analysis. As well as pitfalls. Genomic sequencing using DNA microarrays, RT-qPCR and DNA-seq all have their own sets of pros and cons. Microarrays have sigh sample throughput, but only can identify known genes and transcripts, and can be relatively cost effective. Proteomics-based methods would be able to identify all expressed proteins in the proteome, and identify proteins which are dysregulated based on specific conditions, but can be time consuming due to instrument time. Multi-omics and single-omics methods are significantly more expensive than others due to the specificity. Multi-omics combines multiple omics-based approaches to give a comprehensive understanding of the molecular changes that can contribute to disease state, cellular response and development. Single-cell-omics methods are specialized to reflect only one aspect of a biological system at a single-cell resolution, but is not feasible for large scale samples, thus not being a method to look at multiple disease states. Both single-cell omics and multi-omics methods are high cost, and due to this, small sample sizes are optimal.

All omics methods will have their own pros and cons, as well as optimal sample size, sensitivity and cost, there is no way to compare all to determine which is the best omics methods to use as each method tells a different story about what is happening in the body.

The stage of BC really would depend on what we are looking to get out of the dataset. Are we looking for biomarkers (early onset BC), or molecular mechanisms of metastasis (mid to late stage BC)? This will decide what stages are best for that specific method.

Which omics analysis tools/platforms/websites have the potential to provide more comprehensive information?

Answer: Each omics has their own analysis tools to aid in providing more comprehensive information. In proteomics, we can determine gene names which encode for proteins identified, which can then be used for gene set enrichment analysis (GSEA) and STRING analysis, both of which other omics-based methods can utilize if genes are identified. This is just once example for proteomics experiments, but each will have different tools to get a more comprehensive analysis of their data.

Some of the tools/platforms available online are listed below:

  • BigOmics Analytics (https://www.bigomics.ch), in which the company has an easy to use set of tools called “Omics Analysis for Everyone - Easy-to-use omics tool”.
  • BioCyc (https://biocyc.org/omics.shtml) offers omics data analysis. The website offers multiple tools for analysis of gene expression, metabolomics, and other large-scale datasets. Options for gene expression and metabolomics data are detailed here, but many of the options that involve pathways or the metabolic map can also be used for proteomics, multi-omics, or other kinds of high-throughput data.
  • NetGestalt (https://www.altexsoft.com/blog/omics-data-analysis/) is a web app for multi-omicsdata visualization and integration.
  • MiBiOmics (https://shiny-bird.univ-nantes.fr/app/Mibiomics ) is an interactive web-based (and standalone) application to easily and dynamically explore associations across omicsdatasets
  • Subio Platform (https://www.subioplatform.com/) is professional software for analyzing quantitative omics data like transcriptomics, epigenetics, or proteomics. 

So, there are many options for analysis of multiomics datasets and many of them have the potential to provide more comprehensive information. As we haven’t tested them yet, we are unsure which one of the multiomics software are the best match for multiomics analyzes. However, we can comment on genomics-proteomics-pathway-omics analysis. We always investigate the pathways that are dysregulated in our proteomics experiments, and use the outcome to investigate the gene function using GSEA analysis. We also investigate protein protein interactions (PPIs) and protein post translational modifications (PTMs) using STRING analysis, Reactome, Ingenuity Pathway Analysis, ExPaSy etc.   

Additionally, considering the growing body of evidence suggesting the involvement of the microbiota in breast cancer progression and metastasis, it would be valuable to include a discussion on microbiomics in this review.

Answer: To address this requirement, we have discussed about the microbiome and microbiomics implication within the subchapter 2.5., “Other omics-based investigation of breast cancer”, where we have introduced the following paragraph: Microbiomics is focused on the characterization and quantification of biomolecules responsible for the structure, function, and dynamics of microbial communities [253]. It is known that the breast, gut, and milk microbiomes are involved in occurrence of malignant and non-cancerous breast lesions, suggesting that the microbiome is a valuable risk factor for BC [254], especially through regulation of BC microenvironment [255]. Thus, Zhu et al. demonstrated that the BC in postmenopausal women is associated with an altered gut metagenome [256], while modifications in the microbiota of human milk have consequences for mammary health [257].”